

# Identification and quantification of particulate tracers of exhaust and non-exhaust vehicle emissions

Aurélie Charron[1,2], Lucie Polo-Rehn[1,2], Jean-Luc Besombes[3], Benjamin Golly[3], Christine Buisson[4], Hervé Chanut[5], Nicolas Marchand[6], Géraldine Guillaud[5], and Jean-Luc Jaffrezo[1]

[1]Univ. Grenoble Alpes, CNRS, IRD, IGE (UMR 5001), F-38000 Grenoble, France
[2]IFSTTAR, F-69675 Bron, France
[3]LCME, Univ. Savoie Mont-Blanc, LCME, F-73000 Chambéry, France
[4]LICIT, ENTPE, IFSTTAR, F-69518 Vaux en Velin, France
[5]Atmo Auvergne-Rhône-Alpes, F-69500 Bron, France
[6]Aix Marseille Univ., CNRS, LCE, F-13331 Marseille, France

*Correspondence to*: Aurélie Charron (aurelie.charron@univ-grenoble-alpes.fr)

**Abstract.** In order to identify and quantify key-species associated with non-exhaust emissions and exhaust vehicular emissions a large comprehensive dataset of particulate species has been obtained thanks to simultaneous near-road and urban background measurements coupled with detailed traffic counts and chassis dynamometer measurements of exhaust emissions of a few in-use vehicles well-represented in the French fleet. Elemental Carbon, brake-wear metals (Cu, Fe, Sb, Sn, Mn), n-alkanes (C19-C26), light molecular weight PAHs (Pyrene, Fluoranthene, Anthracene) and two hopanes ($17\alpha 21\beta$Norhopane and $17\alpha 21\beta$hopane) are strongly associated with the road traffic. Traffic-fleet emission factors have been determined for all of them and are consistent with most recent published equivalent data. When possible, light-duty and heavy-duty duty traffic emission factors are also determined. Most of the first ones are in good agreement with emissions from chassis dynamometer measurements in absence of significant non-combustion emissions. This study has shown that ratios involving copper (mainly Cu/Fe and Cu/Sn) could be used to trace brake-wear emissions as they seem to be roughly constant in Europe and as longer as Cu-free brake are not largely spread. In France where the traffic was largely dominated by diesel vehicles in 2011 (70 %), the OC/EC ratio typical of traffic emissions was around 0.44. On the contrary, the use of quantitative data for source apportionment studies is not straightforward for the identified organic molecular markers; while, their presence seems to well-characterized fresh traffic emissions.

## 1 Introduction

Traffic is a major source of particulate matter in urban environment through both exhaust and non-exhaust emissions. Thanks to stringent regulations, vehicles with more efficient catalytic converters and diesel particle filters are progressively introduced into the European fleet. As a consequence particulate exhaust emissions have strongly decreased and non-exhaust particulate emissions (particles resuspended by moving traffic and those from the wear of brakes, tyres, road surface…) will contribute to the major part of particulate vehicular emissions in the near future (Amato et al., 2014). Current research estimates that non-exhaust emissions already substantially contribute to traffic emissions, with differences between sites due to local meteorology, local emissions, or traffic characteristics (e.g. Omstedt et al., 2005; Thorpe and Harrison, 2007; Bukowiecki et al., 2010; Lawrence et al., 2016). It is obvious that even with electric vehicles, traffic will continue to be a source of PM through non-exhaust emissions (Pant and Harrison, 2013; Timmers and Achten, 2016).



Also, the knowledge on the deleterious impacts on human health of PM from vehicular emissions is increasing.
There is now strong evidence that traffic-related PM are responsible for adverse health effects due to the health
effect of both carbonaceous material from exhaust emissions and redox-active metals in traffic-generated dust
including road, brake and tyre wear (Cassee et al., 2013 ; Amato et al., 2014 and references wherein). Recently,
Shirmohammadi et al. (2017) have shown the important role of non-tailpipe emissions to the oxidative potential
of particulate matter species identified as tracers of vehicle abrasion. Therefore, a better knowledge on vehicular
emissions is required to better understand their contribution to urban atmospheric PM concentration levels and
related health effects.
Chassis dynamometer measurements allow the determination of exhaust vehicular emissions under controlled
testing conditions but, because of high costs, these tests often include small sets of vehicles that cannot be
representative of large variation in engine type, age and maintenance history of vehicles. They do not either
represent the variability of driving types in various environments. All of these are parameters that strongly
influence real-world vehicular emissions. Additionally, these tests cannot simulate accurately the effect of dilution
on particle equilibrium in the road atmosphere (Kim et al., 2016), the possible rapid aging effects (e.g. Platt et al.,
2017), and non-tailpipe emissions (e.g. Thorpe and Harrison, 2008). For all these reasons, it is thought that more
realistic estimates of vehicle emissions are determined from air quality measurements in the near-road atmosphere
(Phuleria et al., 2017), including car-chasing, tunnel or roadside measurements (Pant and Harrison, 2013 and
references therein; Jezek et al., 2015).
In this study, a large comprehensive dataset on the chemistry of PM has been obtained thanks to two
complementary campaigns: 1) simultaneous near-road and urban background measurements coupled with detailed
traffic counts for the estimation of real-world vehicular emissions including non-exhaust emissions and, 2) chassis
dynamometer measurements of exhaust emissions of a few in-use vehicles well-represented in the French fleet for
the identification of key species in exhaust emissions. Possible particulate tracers of exhaust and non-exhaust
vehicle emissions are examined and quantified. Emission factors and when possible, typical ratios, are derived
from these data.
**2 Methodology**
**2.1 Chassis dynamometer measurements**
Vehicles were operated on a chassis dynamometer, and exhaust emissions were measured using a sampling train
including a Constant Volume Sampling (CVS) system in which emissions are diluted with filtered air and sampled
using quartz filters and polyurethane foams (PUF).
**2.1.1 Vehicle types and driving cycles**
Five vehicles representative of the most frequent vehicle classes in the French fleet in circulation were selected
(Euro 3 Diesel, Euro 4 diesel, Euro 4 diesel retrofitted with particle filter, Euro 2 petrol, Euro 4 petrol). They were
in-use private cars since rental vehicle may not be representative of the national fleet (low mileage). All vehicles
were operated using commercial diesel and petrol fuels. Selection criteria and characteristics of tested vehicles are
described in the Supplementary Information (SI).
Driving cycles designed to be representative of real driving conditions in Europe are performed in this study
(André, 2004). The ARTEMIS urban cycle represents driving conditions in urban areas (repeated acceleration and



vehicle speed below 40 km/h), while the ARTEMIS rural-road cycle represents driving conditions on main roads
in suburban and rural areas outside cities (flowing traffic conditions). These cycles represent most driving
conditions encountered on the RN87/E712 highway, where the road side experiments took place, during congested
and flowing traffic conditions, respectively. Since urban journeys include cold vehicles, the influence of the cold
start on urban emissions has been systematically tested.
**2.1.2 Exhaust sampling**
Vehicles were operated on a chassis dynamometer using a CVS system to dilute exhaust emissions with filtered
ambient air. The filtration system included four filters and a cartridge in series: a M6-F7-F9, M5, F7 EN-779-2012
filters; a HEPA H13 EN1822-2009 filter; and a cylindrical cartridge of charcoal scrubber.
Regulated emissions were determined with continuous monitors for CO, $CO_2$, $NO_x$, and total hydrocarbons
(HORIBA system). Particulate matter (PM) and volatile organic compounds (VOC) were sampled out from the
dilution tunnel with a servo-controlled system designed by Serv'Instrument for this study. Two driving cycles in
series were performed in order to collect enough matter for molecular speciation (except for the EURO 3 vehicles).
PM was collected on quartz filters (Tissuquartz$^{TM}$, diameter 47 mm), at flow rates depending on both emission
levels of the vehicles being tested and the types of analyses (from 5 to 30 L.min$^{-1}$ for EC/OC measurements and
from 30 to 50 L.min$^{-1}$ for organic molecular speciation). Dilution factors ranged from about 20 to 80. Quartz filters
were pre-baked at 500°C for 8h before being used. Samples were stored at -18°C in aluminum foil and sealed in
polyethylene bags until analyses.
Analysis of test blank filters (collected following the same procedure as vehicle test filters including driving cycle
durations with filtered air) showed contaminations from the CVS system. It was obvious that some organic
compounds measured in test blanks result from the desorption of semi-volatile organics deposited into the CVS,
that is favoured by cleaner air. Since blank levels decreased in time with passing air dilution through the system,
our procedure included such a step maintained for a duration corresponding to at least two driving cycles  before
each vehicle test. Test blanks are used to correct measurements.
**2.2 Near-road and urban background measurements**
The joint PM-DRIVE (PM-Direct and Indirect on-road Vehicular Emissions) and MOCOPO (Measuring and
mOdelling traffic Congestion and POllution) field campaign took place from September 9$^{th}$ to 23$^{rd}$, 2011. It
included meteorology and traffic measurements, near-road/urban background PM$_{10}$ sampling (this study) and near-
road on-line measurements (discussed in DeWitt et al., 2015).
**2.2.1 Description of traffic and urban background sites**
The Grenoble conurbation is a large city with about 700,000 inhabitants. It is located in the southeast of France in
the French Alps and is surrounded by three mountain ranges (Vercors, Chartreuse, and Belledone). The traffic site
was located (45.150641 N, 5.726028 E) about 15 m away from the southern part of the Grenoble ring road (the
RN87/E712 highway with 2×2 lanes) (26 m from the roadway central axis). Total traffic flow for the 4 lanes was
on average 95,000 and 65,000 vehicles a day on workday and weekend, respectively, with frequent congestion
during extended commuting hours. Stop-and-go traffic in both directions generates large braking activity at that





time. This sampling site was highly equipped during the joint PM-DRIVE / MOCOPO field campaign.
Simultaneous measurements took place at an urban background site (Les Fresnes) which is located about 2 km
away from the traffic site, on the rooftop of a school. This site belongs to the regional air pollution network
managed by Atmo Auvergne-Rhône-Alpes (http://www.atmo-auvergnerhonealpes.fr/). The locations of both sites
are presented on Figure 1.
**2.2.2 Aerosol and gas measurements**
On both traffic and urban background sites, $PM_{10}$ HiVol (DA80, 30 $m^3$ $h^{-1}$) samplers collected $PM_{10}$ on quartz
fibre filters (Tissuquartz$^{TM}$, diameter 150 mm) with a time resolution of 4 hours. Filters were preheated at 500 °C
during 3 h. After sampling, filters were individually placed in aluminum foil, sealed in polyethylene bags and
stored at −18 °C until analysis. In addition, both sites were equipped with $NO_x$ (NO and $NO_2$) and CO monitors;
$PM_{10}$ and $PM_{2.5}$ mass concentrations were also continuously measured using TEOM-FDMS.
**2.2.3 Traffic measurements**
Traffic counters (double electromagnetic loops) were installed in order to identify the passing of all vehicles, the
length of their chassis and their speeds,  the determination of the 2 vehicle classes used in this study (light-duty
and heavy-duty vehicles), and the identification of periods of stop-and-go or flowing traffic. Vehicle mean speeds
are computed as the harmonic mean speed of the cars passing over the detector (flow speed i.e. the spatial mean
speed) (Hall, 2001).
Traffic cameras mounted on a roadway gantry were also used to monitor traffic at the measurement site. They
were used to capture the license plate numbers of passing vehicles. Plate numbers were later used to classify
vehicular traffic into different categories: EURO standards and fuel type (diesel or petrol). The traffic of the
highway is detailed elsewhere (DeWitt et al., 2015; Fallah Shorshani et al., 2015). On weekdays, the average
hourly traffic included 2,850 diesel vehicles (including about 200 heavy-duty vehicles) and 1,025 petrol vehicles.
The harmonic vehicle mean speed was about 80 km/h and ranged from 52 to 94 km/h (note that the RN87 highway
has a speed limit of 90 km/h). A sample is considered affected by congested traffic above a threshold of more
22,000 vehicles during the 4-h sampling periods (corresponding to vehicle speed below 70 km/h) (see SI file).
However, braking may occur before the installation of congestion.
The vehicle fleet was close to the national one for the year 2011, with 72% diesel vehicles, and EURO 3 and 4
vehicles representing most of the vehicles (respectively 30 and 36% respectively). Virtually all heavy-duty vehicles
are diesel.
**2.2.4 Meteorological measurements**
A Young meteorological station was installed at the traffic site to capture wind speed and direction, while relative
humidity, and temperature data and rain data are obtained from two stations located in Grenoble conurbation (see
SI file).
The wind speed was low during the field campaign (on average 0.98 m/s, ranging from 0.50 to 2.5 m/s). The
temperature was warm (on average 17.6°C, peaked to 27°C) and was higher before the first rain event (on Sunday
17/09/11) with average temperatures of 21°C and 15°C, respectively, before and after the rain event. Two rain





events occurred during the campaign, the first one from the morning of 17/09/11 to the night of 18/09/11 night (34
mm of rain) and the second one from the night of 18/09 to 19/09/11 afternoon (12.7 mm). They corresponded to
summer storms.

### 2.3 Chemical analyses

The measurements of carbonaceous material (EC and OC) in PM samples were performed using the Thermo-
Optical Transmission (TOT) method on a Sunset Lab analyser (Jaffrezo et al., 2005; Aymoz et al., 2006) following
the EUSAAR2 temperature protocol (Cavalli et al., 2010). Ionic species were analyzed with Ionic Chromatography
(IC) following a well-established method (Jaffrezo et al., 1998; Waked et al., 2014). Metals were analyzed using
Inductively Coupled Plasma Spectrometry (ICP-MS) (Waked et al., 2014). The chemical speciation of organic
particles are performed by Gas Chromatography–Mass Spectrometry (GC-MS) and liquid chromatography
(HPLC) using a fluorescence detector (Piot, 2011; Golly et al., 2015).
Hence, a large number of particulate chemical species have been measured in the filter sampler simultaneously
collected at the two traffic and urban background sites, including EC, OC, 9 major ions (Cl⁻, $NO_3^-$, $SO_4^{2-}$, oxalate,
Na⁺, $NH_4^+$, K⁺, $Mg^{2+}$, $Ca^{2+}$), 33 metals and trace elements (Al, As, Ba, Ca, Cd, Ce, Co, Cr, Cs, Cu, Fe, K, La, Li,
Mg, Mn, Mo, Na, Ni, Pb, Pd, Pt, Rb, Sb, Sc, Se, Sn, Sr, Ti, Tl, V, Zn, Zr), 3 sugars (galactosan, mannosan,
levoglucosan), 15 PAHs (phenanthrene, anthracene, fluoranthene, pyrene, retene, benzo(a)anthracene, chrysene,
benzo(e)pyrene, benzo(b)fluoranthene, benzo(k)fluoranthene, benzo(a)pyrene, benzo(ghi)perylene,
dibenzo(a,h)anthracene, indeno(1,2,3-cd)pyrene, coronene), 30 n-alkanes (from C10 to C40), 2 branched alkanes
(pristane and phytane) and 10 hopanes (trisnorneohopane, 17α-trisnorneohopane, 17α,21β-norhopane, 17α21β-
hopane, 17α,21β,-22S-homohopane, 17α,21β,-22R-homohopane, 17α,21β,-22R-bishomohopane, 17α,21β,-22S-
bishomohopane, 17α,21β,-22S-trishomohopane, 17α,21β,-22R-trishomohopane).
The same array of chemical species were also quantified in the filter samples from the chassis dynamometer
experiments.

### 2.4 Data analysis

A series of multivariate data analysis tools have been used in order to define which species are related to traffic,
to identify influential parameters, and to quantify their respective influences. Thanks to simultaneous
measurements at near-traffic and background sites, incremental concentrations at the traffic site have been
calculated as the difference between near-traffic and urban background concentrations. Sign and Rank Signed-
Wilcoxon tests have been used to estimate if concentrations measured at the near-traffic site are significantly higher
than concentrations measured at the urban background site and can possibly be ascribed to local traffic emissions.
As a complementary indication of relation to traffic, Spearman correlations with traffic data (total traffic, light-
duty traffic, heavy-duty traffic), with $NO_x$ (as an indicator of traffic emissions) and with EC (as an indicator of
diesel traffic emissions) (Morawska and Zhang, 2002; Reche et al., 2011) have also been examined. Results for
species for which concentrations are significantly higher at the near-traffic site (then including both positive
increments and p-values below 0.05 from Sign and Signed Wilcoxon tests) are presented in Tables 1a and 1b.
Incremental concentrations for all species strongly associated with traffic are transformed into emissions according
to a well-established procedure (e.g. Pant and Harrison, 2013; Charron and Harrison, 2005; details in the SI section





V). This method enables the calculation of average emission factors for the mixed traffic fleet of the RN87 highway
assuming that (1) incremental concentrations are from local traffic, (2) emissions of $NO_x$ are known and estimated
from emission inventories (COPCETE that uses emission functions from European COPERT4 averaged for fleet
composition and speed) and detailed traffic counts and (3) atmospheric dilution affects similarly all pollutants.
Average emission factors (EFs) are expressed in $mg.veh^{-1}.km^{-1}$ or $\mu g.veh^{-1}.km^{-1}$ (Tables 2a,b,c,d). Since it is
thought that nearby industrial activities influence the concentrations of Ti, Cr, Fe and Mn in the morning, these
data are excluded from the calculations. Multiple Linear Regression analyses are performed between emission data
and light-duty and heavy-duty vehicle counts. The coefficients of the regressions represent average EFs for local
heavy-duty and light duty traffics. The constants represent the part not related to local traffic. Results from Multiple
Linear Regressions are not available for all species strongly related to traffic since its use requires normally-
distributed data and the model must be validated through residues analyses.
**3 Results**
Average concentrations measured at the traffic site are presented in the SI (section II). Four-hour $PM_{10}$
concentrations ranged from 4.9 to 45.1 $\mu g.m^{-3}$ (on average 24.1 $\mu g.m^{-3}$) and was strongly dominated by EC (on
average 5.9 $\mu g.m^{-3}$) and OC (on average 5.4 $\mu g.m^{-3}$). At this site in the vicinity of traffic dominated by diesel
vehicles, EC and OC concentrations were of similar magnitude. EC concentrations peaked at commuting time in
the early morning and at the end of the afternoon, and follows quite well the evolution of traffic; the temporal
evolution of OC is closer to that of sulphate with somewhat stronger variability following those of traffic. The
other major constituents of $PM_{10}$ were sulphate (on average 1.2 $\mu g.m^{-3}$), iron (on average 1.0 $\mu g.m^{-3}$) and calcium
(on average 0.8 $\mu g.m^{-3}$). Strong showers during the weekend in the middle of the two-week campaign led to $PM_{10}$
concentrations below 10 $\mu g/m^3$during a few hours (see SI VI for individual particulate species).
**3.1 Identification of species related to local on-road traffic**
Tables 1a and 1b present species for which concentrations are significantly higher at the near-traffic site and
assumed from local emissions. These species could be distributed into two main groups according to both the
significance of the contribution of traffic to atmospheric levels and the strength of relations with traffic indicators
(traffic counts, $NO_x$ and EC concentrations).
**3.1.1 Species strongly associated with local traffic**
Species significantly correlated with traffic indicators and for which local traffic contributions are above 50% are
highlighted in Tables 1a, 1b.
Cu, Fe, Mn, Sb, Sn have concentrations strongly correlated with traffic indicators (Table 1a). Their incremental
concentrations range from 54% to 84%. The concentrations of these species are linearly related to each other with
near-zero intercepts (SI, section VIII) confirming that they come from the same source. It should be noted that Fe
and Mn present an additional source at this site in the morning, as discussed in section 3.2. Similarly to this study,
Amato et al. (2011a) and Harrison et al. (2012) measured strong incremental concentrations for Fe, Cu, Sb and Sn
at traffic sites with strong correlations between them. Here, Cu, Fe and Sn are the metals that are the most closely



related, while relationships with Mn and Sb are more scattered and possibly more closely related to the heavy-duty
traffic (Table 1a and SI).
Cr and Ti are also species significantly correlated to traffic indicators but to a lower extent than Cu, Fe, Mn, Sb
and Sn. Cr (Boogaard et al., 2011 ; Amato et al., 2011a) and Ti (Amato et al., 2011a) also showed higher
atmospheric concentrations at street locations than at urban background sites in previous studies. Cr and Ti show
a behaviour different from that of other elements coming from brake wear. They present sharp peaks in the
morning, poorer correlations with copper, and significant correlations with Al (see SI sections VII and Table 1a).
When the very high morning concentrations are removed (mornings of workdays), their temporal variations are
much closer to the ones of metals from brake wear emissions. This suggests that another source influenced Cr and
Ti concentrations near the traffic site in the morning, possibly nearby metalworking activities. Similarly, while
morning peaks are less obvious in the temporal variations, the exclusion of morning data for Fe and Mn improves
their relationships with Cu (see SI section VIII).
Many of these species are metals that are known to arise from brake wear emissions (Thorpe and Harrison, 2008;
Pant and Harrison, 2013; Grigoratos and Martini, 2014, 2015). Indeed, Fe could come from the lining (steel or
iron powder) in semi-metallic brake, from fibres as steel, or cast iron rotor wear for low metallic brake. Cu is a
high-temperature lubricant present in linings and it is also included in fibres as brass to increase braking
performance. Sb is an element of brake lining both in filler as antimony sulphate and in lubricant as antimony
trisulphide. Chromium oxides are elements of the filler of brake linings used for their thermal properties, and
potassium titanate fibers are present as a strengthener in organic linings (Sanders et al., 2003; Grigoratos and
Martini, 2014 and references therein). Cr could also come from lubricant oil combustion (Pulles et al., 2012).
Only three light molecular weight PAHs (An, Fla, Pyr) are strongly associated with traffic indicators even though
only the particulate phase is determined (Table 1b). It is well-established that light molecular weight PAHs are
emitted by diesel vehicles, while higher molecular weight PAHs are rather associated with petrol vehicle emissions
(Zielinska et al., 2004; Phuleria et al., 2006 and 2007; Pant and Harrison, 2013). Not surprisingly, in this site
dominated by diesel vehicles, high molecular weight PAHs (from 5 rings) are more significantly correlated with
levoglucosan, suggesting a closest relation to biomass burning emissions than to on-road petrol vehicles.
Proportional concentrations of An, Pyr, and Fla suggest that they mainly come from the same source, likely diesel
exhaust emissions (SI section VII). Specifically, Pyr and Fla show high incremental concentrations (as an
indication of the low contribution of the urban background compared to strong traffic contribution, and/or rapid
photochemical degradation) and strong linear relationship without any significant intercept (as an indication of
common origin). Concentrations of An have more variability and larger contribution from the urban background.
The n-alkanes C19 to C26 and two hopanes ($17\alpha21\beta$norhopane and $17\alpha21\beta$hopane) are species that are not
significantly, or only weakly, correlated to total traffic. However, they are significantly correlated with the heavy-
duty traffic, $NO_x$ or EC. The n-alkanes between C18 and C25 are the predominant ones in vehicle exhaust and
correspond to the high boiling point components in diesel fuels (Alves et al., 2016). Hopanes are known to arise
from unburnt lubricating oil emissions (Rogge et al. 1993a; Zielinska et al., 2004), and $17\alpha21n$Norhopane is
considered as a tracer for lubricating oil (Kleeman et al., 2008). While n-alkanes have been measured in the exhaust
of test diesel vehicles in our study (Table 4), hopanes and steranes were below the quantification limit for all
selected vehicles. It was assumed that the tested vehicles (passenger cars) did not emit enough unburnt oil for the
proper quantification of hopanes. At the sampling site, the temporal variations of the concentrations of these



species show higher values on the second week of the field campaign than on the first one. This observation can
be explained by much larger heavy-duty traffic during this period (almost twice the traffic of the first week).

### 3  3.1.2 Other species

This second group corresponds to species with local increments below 50% and no significant, or only weak,
correlations with traffic indicators. It is highly heterogeneous since it includes OC, $NO_3^-$, $Na^+$, $Ca^{2+}$, $Mg^{2+}$, Ba, Co,
Phenanthrene (Phe), Benzo(a)Anthracene (BaA) and some n-alkanes (C18, and from C27 to C33).
Not surprisingly $NO_3^-$ is not significantly related to primary traffic indicators. However its concentrations are
significantly higher at the traffic site ($p<0.001$, and median increment of 28%). Additionally, the temporal pattern
of $NO_3^-$ is different from that of indicators of regional origin (for example $SO_4^{2-}$, SI section VI). Amato et al.
(2011a) found similar contributions of traffic to $NO_3^-$ concentrations at two traffic sites in Barcelona (34% and
25% respectively). They assumed that $NH_4NO_3$ is quickly formed in the road plume enriched in $NH_3$ emitted by
vehicles.  Indeed, the important introduction of vehicles equipped with DeNOx technology such as 3-way catalysts
or selective catalytic reduction systems have led to recent increasing vehicular $NH_3$ emissions (Kean et al., 2009;
Suarez-Bertoa and Astorga, 2016) responsible for significant changes in the chemistry of the atmosphere in
roadside areas.
OC concentrations are significantly higher at the traffic site ($p<0.001$; increment of 23%), and are weakly but
significantly correlated with traffic indicators. Amato et al., (2011a) found slightly higher increments of 34% and
41% for OC at their 2 traffic sites. Deconvolution of sources from simultaneous $PM_1$ AMS measurements
concluded that about 20% of the total $PM_1$ organic mass could be attributed to vehicular emissions (DeWitt et al.,
2015). Additionally, the temporal pattern of OC concentrations clearly shows a dominant regional contribution (SI
section VI). All these results agree that in summer time, the majority of OC is of regional origin, despite a clear
influence of traffic.
$Na^+$, $Ca^{2+}$, $Mg^{2+}$ and Sr concentrations are also significantly higher at the traffic site ($p<0.05$, increments from 33%
to 43%, except for Sr for which urban background concentrations are most of the time below detection limit).
When three high concentrations for $Mg^{2+}$ concentrations and one for Sr concentrations are excluded (all probably
not related to traffic), both present temporal variations close that of calcium (SI section VII). All are at least
significantly correlated to the heavy-duty traffic ($p<0.05$). Since $Ca^{2+}$, $Mg^{2+}$ and Sr are strongly affected by rain
events (assumed to influence the silt loading of the road), their incremental concentrations are believed to be from
resuspended dust, as concluded in other studies (e.g. Lough et al., 2005, Amato al., 2011a).
Ba and Co concentrations are significantly higher at the traffic site ($p<0.05$, increments of 37% and 36%
respectively). Ba concentrations present a temporal behaviour close to the ones of Cu, Fe, Sn and Mn and,
accordingly, is known as an element of the filler of brake linings. Higher Ba atmospheric concentrations at street
locations than at urban background sites have also been observed in previous studies (Harrison et al., 2012). On
the contrary, Co concentrations do not show any correlation with traffic indicators and are correlated with tracers
of regional origin (Table 1a). Co is sometimes measured in brake pads (e.g. Hulskotte et al., 2014); and Amato et
al. (2011) found similar increments for Co at their traffic sites (respectively 33% and 47%). All of these suggests
a possible contribution of brake wear to Co incremental concentrations, despite the lack of correlation with traffic
indicators.



The concentrations of Phe, BaA, C18 and C27-C33 alkanes are significantly higher at the traffic site and some of
them are significantly correlated with $NO_x$ and EC. Accordingly these species were detected in the exhaust
emissions of diesel vehicles sampled from our chassis dynamometer experiments. However most of them are also
significantly correlated with levoglucosan measured at the traffic site, particularly BaA. Indeed the temporal
pattern of BaA shows similarities with both the one of Pyr (strongly related to traffic, see below) and levoglugosan
(tracer of biomass burning peaking in the evening) (SI section VI). It is therefore believed that most of them are
largely emitted by biomass burning and are not specific of road traffic emissions.
Most of species from this group are obviously emitted by the traffic but they seem to be less specific than species
from the first group since other sources can largely contribute to their emission levels.

**3.2 Quantification of traffic fleet emissions for species associated with traffic**

Average emission factors (EFs) are presented in Tables 2a,b,c,d. Most EFs show large standard deviations. This
variability reflects the presence of vehicles with various emission levels (diesel/petrol; different standards and
engine load; cold start/hot vehicles). It could also be related to the variability of the vehicle fleet (on-average 5%
heavy vehicles but ranging from 0.3% to 12%) and to the various traffic conditions (from fluid with speeds up to
90 km/h to congested with stop-and-go traffic).
Table 3 presents average EFs for heavy-duty and light duty traffics, their standard deviations and confidence
intervals at 95%. All constants are not significant (p-values above 0.4 for metals and organics, p-value of 0.061
for EC) suggesting that mostly local traffic contribute to local incremental concentrations.

**3.2.1 EC/OC emissions**

The traffic-fleet EF for EC determined in this study (39 mg.veh$^{-1}$.km$^{-1}$) is slightly higher than the ones determined
in tunnel studies in Austria, China, California, and near a heavily-trafficked highway in Switzerland (about 21
mg.veh$^{-1}$.km$^{-1}$ for Handler et al., 2008; Hueglin et al., 2006; Cui et al., 2016; and 31 mg.veh$^{-1}$.km$^{-1}$ for Gillies et
al., 2001), while it is similar to the one of a tunnel study in Portugal (39 mg.veh$^{-1}$.km$^{-1}$) (Alves et al., 2015).
Conversely, the traffic-fleet EFs for OC in Austria and China (about 19 mg.veh$^{-1}$.km$^{-1}$) are similar to ours, while
the ones for urban tunnels in Portugal (Alves et al., 2015) and U.S. (Gillies et al., 2001) are higher (39 mg.veh$^{-1}$.km$^{-1}$
.km$^{-1}$ and 25 mg.veh$^{-1}$.km$^{-1}$ respectively). Different traffic fleet and traffic conditions may explain these small
divergences. Part of the differences may also be due to the different measurement technics used for the distinction
between EC and OC.
The average light-duty traffic-fleet EF for EC is in excellent agreement with the EFs of Euro 3 and Euro 4 diesel
vehicles obtained from chassis dynamometer measurements in our study for which the highest EFs were clearly
observed when vehicles are cold (Figure 2a). Indeed, these two types of vehicles represented the largest proportion
of passenger's cars on the RN87 freeway in 2011 (Fallah Shorshani et al., 2015), as well as in the French national
fleet. Similarly to results for diesel vehicles tested by Fujita et al. (2007) and Lough et al. (2007), EC has the
highest emission factor in diesel exhaust. It also could be noted that the heavy-duty traffic-fleet EF for EC is about
5-times higher than the one determined for light-duty traffic.
The emissions of OC could not be discriminated between light-duty and heavy-duty traffic (coefficients not
significantly different from zero). However, it can be observed that the traffic-fleet EF for OC is larger than it
could be expected from exhaust measurement of test vehicles (Table 4, Figure 2b). Note that traffic-fleet EFs for





OC from other studies (Handler et al., 2008; Alves et al., 2015, Cui et al., 2016; He et al., 2008) are at least as high
as ours and EFs for exhaust OC from Cheung et al. (2010) for Euro 4 diesel vehicles (with and without DPF) is
similar to ours. There are many likely explanations for the traffic-fleet EF for OC being higher than expected:
proportionally larger contribution of the heavy-duty traffic to OC than to the EC; contribution of non-exhaust
emissions to the OC (e.g. tyres wear); contribution of smoker vehicles, rapid formation of secondary OC in the
roadside atmosphere. Further studies are required to assess the respective importance of these processes.
Average OC/EC ratios on incremental concentrations and emissions are respectively 0.33 and 0.44 (Table 5) that
is in excellent agreement with observations in a roadway tunnel in Lisbon and in roadside incremental
concentrations in Birmingham, U.K. (Pio et al., 2011; Alves et al., 2016). Pio et al. concluded that OC/EC ratios
of 0.3-0.4 characterizes vehicle fuel combustion at their sites and similar conclusions could be made for Grenoble.
However, the OC/EC ratio is expected to depend on vehicle fleet. Indeed, in the exhaust of test diesel vehicles
non-retrofitted with particle filters (Table 4), OC/EC ratios are lower than 0.4, while they are respectively 6.1 and
1.7 in the exhaust of Euro 2 petrol vehicle (urban cold cycle) and Euro 4 petrol vehicle (road cycle) (no average
OC/EC ratios has been determined for petrol vehicles because of many measurements below detection limit). Other
recent chassis dynamometer measurements (Lough et al., 2007) showed that most OC/EC ratios are above 2 for
petrol vehicles (except for two smoker vehicles) and below 0.5 for diesel vehicles (except idling conditions), with
even lower ratios for older and heavier diesel vehicles. Because in 2011, diesel vehicles represented about 70% of
total vehicles on the RN87 highway, the OC/EC ratio found in this study is lower than the ones of many other
studies. For examples, OC/EC ratios from EFs or incremental concentrations were 0.90±0.21 in a tunnel in Austria
(Handler et al., 2008); on average 0.52 for a traffic fleet with 40% diesel vehicles in Spain (Amato et al., 2011a)
and 0.5 in Marseille where petrol two-wheelers are much more numerous (El Haddad et al., 2009). However, since
the OC/EC ratio in the exhaust of the diesel vehicle equipped with particle filter was higher (0.7) possibly due to
the more efficient trapping of EC than OC by the particle filter, one can expect that in the future the OC/EC ratio
reflecting vehicle fuel combustion would increase with the progressive introduction of vehicles equipped with PF.
**3.2.2 Metals from brake wear emissions**
Fe presents by far the third higher traffic emission rate after EC and OC (6.7 mg.veh$^{-1}$.km$^{-1}$, Tables 2). Cu also
shows a high emission factor from traffic (300 µg. veh$^{-1}$.km$^{-1}$ respectively). Similarly, major contributions of Fe
and Cu to PM$_{10}$ vehicular emissions were found in other locations (e.g. Kam et al., 2012; Harrison et al., 2012).
The analysis of brake wear debris (Apeagyei et al., 2011 ; Sanders et al., 2003 ; Kukutschová et al., 2009 ; Hulskotte
et al., 2014) and brake material (Gigoratos and Martin, 2015; Hulskotte et al., 2014) confirmed that Fe is the major
element of brakes and that Cu is another important element. Indeed, maximum contents of 60% by mass for Fe
(Gigoratos and Martin, 2015) and of 15% for Cu (Denier van der Gon et al., 2007) have been reported for brake
lining materials. Fe, Cu, Zn, Sn would represent about 80-90% of metals presents in brake pads and Fe 95% of
metals in brake discs (Hulskotte et al., 2014)
Our results can be compared with other European traffic EFs determined for PM$_{10}$ species measured in roadside
environments or tunnels, and with data derived from average brake wear composition (details in SI, section IX).
Most PM$_{10}$-EFs are consistent with other roadside traffic fleet emissions (Johansson et al., 2009; Bukowiecki et
al., 2009) and with data corresponding to braking emissions of 8 mg.veh$^{-1}$.km$^{-1}$ (Hulskotte et al. 2014), but larger
than the estimations aimed to quantify brake wear only (Bukowiecki et al., 2009). Our traffic-fleet EF for Sb is



much lower than similar EFs determined in other studies, while Ba emissions show large variability from a study
to another. EFs related to brake wear component emissions are lower in tunnel environments (Handler et al., 2008;
Alves et al., 2015), except for the oldest study (Gillies et al., 2001). This could be explained by less stop and go
traffic in such environments. Indeed, interestingly, many EFs determined by Handler et al. (2008) are about half
the ones found in this study. Even though this proportionality is not found for Ba, Sb and Ti, this supports that
brake wear metal EFs are strongly related with braking strength. The low Sb emissions of our study may possibly
be related to the introduction of Sb-free brake pads (von Uexküll et al., 2005 ; Apeagyei et al., 2011).
EFs for brake wear metals (Cu, Fe, Sb and Sn) are 8 to 13 times higher for the heavy-duty traffic than for the light-
duty traffic (Table 3). The estimations from Bukowiecki et al. (2009) for brake wear only are much lower than
ours but somewhat proportional (factors 4.6 to 6.8 for light-duty EFs and from 4.3 to 8.5 for the heavy-duty traffic
– Sb excluded). This consistence between brake profiles suggests that brake compositions would be similar in
different European countries. Traffic-fleet light-duty EFs for brake wear metals are much higher than the ones
from chassis dynamometer exhaust measurements (this study: Table 4; Cheung et al., 2010), confirming the
dominant contribution of braking for these elements.
The sum of traffic-fleet EFs for metals related to brake wear (Ba, Cr, Cu, Fe, Mn, Sb, Sn, Ti - Table 2b) leads to a
total of 7.3 mg/km that should be lower than the real emission factors for brake wear dusts since it only includes
metals, and does not take into account of carbonaceous material and other unquantified compounds of brakes (S,
Zn, Al, and Si, for the most important ones). Hulskotte et al. (2014) determined a brake profile from the analysis
of 65 brake pads and 15 brake discs from 8 very common car brands in Europe. The consistence between with
their data and ours suggests that brakes spent in France are quite similar to the ones spent in The Netherlands.
Then, their average brake profile data are used to estimate the average emission factor for brake wear assuming
that the total proportion of metals is kept, and, according to Hulskotte et al., that 70% of the wear arise from the
disc. Since exhaust emissions (chassis dynamometer measurements or EFs from COPCETE inventory when
available) represent less than 5% of total emissions for Cr, Cu, Fe, Mn, Sb, Sn, and Ti, traffic-fleet EFs for these
elements are assumed to be entirely due to the emissions of brake wear dusts. Then, including unquantified
compounds leads to the rough estimation of 9.2 mg/km for emissions related to brake wear at the RN87 highway.
This estimation includes particles directly emitted during braking and those resuspended, and emissions from light
and heavy-duty traffic. It should be highlighted that this EFs for brake wear emissions is almost twice the particle
emission standards for the exhausts of Euro 5 and Euro 6 vehicles (5 mg/km).
Atmospheric copper is obviously largely from brake wear. Indeed brake wear represents 50-75% of European Cu
emissions (Denier Van der Gon et al., 2007) and 64% of French Cu emissions (CITEPA emission inventory,
citepa.org/fr/air-et-climate/polluants/metaux-lourds/cuivre). Since Sb is another well-known constituent of brake
with very few other atmospheric sources, Cu/Sb ratio is a candidate to trace brake wear emissions (Gietl et al.,
2010; Pant and Harrison, 2013). Pant and Harrison (2013) reviewed Cu/Sb ratios in brake wear particles and found
ratios from 1.3 to 9.1 that strongly depends on the PM fraction. The estimation for the Cu/Sb ratio (11.7±5.1 for
incremental $PM_{10}$) is close to the ones for $PM_{10}$ elemental concentrations in other European locations including
London, U.K. (on average 9.1, Gietl et al., 2010), Barcelona, Spain (7.0-7.9, Amato et al., 2011a); Bern and Zurich,
Switzerland (respectively 13.2 and 8.5, Hueglin et al., 2005) and in road dust below 10 µm collected in Barcelona,
Zürich and Girona (respectively 6.8±0.9, 13.5±6.1, 17.0±8.9, Amato et al., 2011). However, lower $PM_{10}$ Cu/Sb
ratio are found for other European traffic sites, in Sweden (4.6±2.3, Sternbeck et al., 2002 and 3.8, Johansson et




al., 2009); in Portugal (about 2, Alves et al., 2015) and in Vienna, Austria (1.6, Handler et al., 2008). To complicate
the matter further, published data from the analysis of brake pads and discs are even more scattered, roughly from
1.3 to 2000 (Adachi et al., 2004; Canepari et al., 2008; Ijima et al., 2007), and often different than the chemical
composition of brake wear particles (Grigoratos and Martini, 2015). The poorer relationship between Cu and Sb
($r^2$= 0.4) than those observed between Sn, Cu and Fe ($r^2$>0.7) and the lower EF for Sb of this study than
observations at other sites may be explained by the introduction of Sb-free brake pads. Then, while the occurrence
of Sb may be largely explained by braking activities, the use of a typical factor for braking activities using Sb
seems to be less obvious.
The very strong linear relationships found between Fe, Cu, Mn and Sn with no significant intercept (virtually equal
to zero) suggests that ratios including these compounds also worth attention. Even though Fe is much less specific
of brake wear emissions than Cu or Sb, Hulskotte et al. (2014) observed a stable Cu/Fe ratio of about 4 % in
different European kerbside locations (The Netherlands: Boogaard et al., 2011; England: Gietl et al., 2010;
Switzerland: Hueglin et al., 2005). They concluded that brake wear is probably the single source of both copper
and iron in urban aerosols. The review of other recent studies leads to similar conclusions. Pio et al. (2013)
determined an average Fe/Cu ratio of 21 (and then, Cu/Fe = 0.048) from tunnel and busy roads measurements in
Portugal. Cu/Fe ratios at two traffic sites in Barcelona ranged from 0.034 to 0.058 (Amato et al., 2011a). Except
for the site of Weerdsingel in Utrecht, Cu/Fe ratios calculated from average $PM_{10}$ elemental concentrations
measured at 8 street locations in The Netherlands also ranged from 0.039 to 0.048 (Boogaard et al., 2015). Only
Alves et al. (2015) found a much higher Cu/Fe ratio in $PM_{10}$ collected in a tunnel in Portugal mainly due to a very
low EF for Cu compared to other studies. The Cu/Fe ratio determined in our study for light-duty traffic
(0.041±0.010) is very similar to the ones found in the majority of these European roadside sites (Table 5), while
the ratio determined for the heavy-duty traffic (0.070±0.017) is slightly larger due to proportionally higher
emissions of Cu. Again, this suggests that brake materials spent in different European countries are very similar
and that the Cu/Fe ratio could be used as an indication of brake wear emissions.
Ratios with Mn and Sn are more rarely discussed in the literature, and published information is scarce. Cu/Sn ratios
of 6.2 and 4.3 can be calculated from data published by Handler et al. (2008) and Johansson et al. (2009),
respectively.  Amato et al. (2011a) found Cu/Sn ratio that ranged from 4.7 to 4.8 and did not estimate ratios with
Mn because of the possible influence of a local steel source. Cu/Mn ratios of 3.7; 4.9 and 3.0 (Bern) and 4.4
(Zurich) are respectively calculated from the studies of Handler et al. (2008), Johansson et al. (2009), and Hueglin
et al. (2005). All these ratios are close to the ones of this study (Cu/Sn=4.1±1.0; Cu/Mn=3.6±1.6, Table 5), showing
that they are possible other candidates to trace brake wear emissions. Cu/Sn would be a better candidate since the
relation between these 2 elements is closer ($r^2$=0.84 for Cu/Sn vs. $r^2$=0.47 for Cu/Mn, see SI).
**3.2.3 Emissions of organic markers of diesel exhaust and lubricating oil**
As previously observed (Schauer et al., 2002; Perrone et al., 2014; El Haddad et al., 2009), n-alkanes are abundant
organic compounds of total quantified organic compounds emitted by vehicles on road.
In agreement with other chassis dynamometer measurements (Rogge et al.; 1993a; Cheung et al., 2010; Perrone et
al., 2014; Cui et al., 2017), the most important n-alkanes in diesel exhaust emissions are C19 to C26, while, most
emissions were below detection limit for the two petrol vehicles (Table 4). Schauer et al. (2002) showed that n-
alkanes are present in particulate exhaust emissions of non-catalyst equipped petrol vehicles, but they are almost



absent in the exhaust of catalyst-equipped petrol motor vehicles. In 2011 virtually all petrol vehicles were catalyst-
equipped (Fallah Shorshani et al., 2015). C19-C26 n-alkanes are also the most important in incremental roadside
concentrations and as a further indication of road traffic contribution, the Carbon Preference Index (CPI, for
calculation see SI section VI) is very close to unity (0.99) indicating equal distribution between odd and even
carbon number typical of anthropogenic emissions (Harrad et al., 2003; Cincinelli et al., 2007 ; Kam et al., 2012;
Alves et al., 2016). Light-duty traffic-EFs for C23 and C24 are in agreement with the ones of the Euro 3 diesel
vehicle from chassis dynamometer measurements (upper range) and with the ones determined by Perrone et al.
(2014) for Euro 3 diesel vehicles (C23: 16.3 µg/km ; C24: 12.7 µg/km). Good agreements are also found for C25
and C26. However, the traffic-EFs determined by Perrone et al. for C21 is half the one of this study, the one for
C20 is 1/5 of the one of this study. All EFs determined by Perrone et al. for Euro 4 diesel vehicles are larger the
ones of the Euro 4 diesel vehicle of this study, but of similar size of magnitude.
The relative contributions of the n-alkane series in diesel exhaust show Gaussian-like shapes peaking between C20
and C22 (Rogge et al.,1993a; Morawska and Zhang, 2002; El Haddad et al.,2009; Cheung et al., 2010; Fujitani et
al., 2012; Perrone et al., 2014; Cui et al., 2017). In this study, the dominant alkanes are C21 in the exhaust of test
diesel vehicles and C22 in traffic RN87 emissions (Figure 3). Fujitani et al. (2012) showed that shifts from C20 to
C22 and differences between real-atmosphere measurements and exhaust measurements are entirely explained by
gas-to-particle partitioning at different dilution ratios (dominant n-alkanes were C20 and C22 at dilution ratios of
14 and 238 respectively). Accordingly, in this study, C21 is the dominant n-alkane at diesel exhaust dilution ratios
of about 80; and a shift toward a heavier n-alkane has been observed in the roadside atmosphere. N-alkanes
normalized by OC (µg/g) (Figure 3) show values for Euro 3 and Euro 4 diesel emissions and RN87 traffic
emissions that are about half of the values measured in a tunnel in Marseille (El Haddad et al., 2009). Much higher
values are found with the vehicle retrofitted with PF (Figure 3). These high values are due to very low OC
emissions by the vehicle equipped with PF and clearly show less effective reduction of n-alkanes than total OC.
Because of their semi-volatile nature, n-alkanes may pass the particle filter in their gaseous phase at higher
temperatures. While the n-alkanes series from C19 to C26 peaking around C20-C22 seems to be characteristics of
diesel emissions, increase of the n-alkane/OC values could be expected for the future.
Low molecular weight PAH's (An, Fla, Pyr) are frequently associated with particulate diesel exhaust emissions
(e.g. Kleeman et al., 2008; Keyte et al., 2016). Accordingly, both chassis dynamometer and *in-situ* measurements
indicate that diesel vehicles is a major source of these chemical species. Indeed, they are almost absent in petrol
exhaust emissions and are strongly related to RN87 traffic emissions. EFs for PAHs determined for the heavy-
duty traffic are much higher than the ones determined for light-duty traffic: from 6-times higher for Fla, up to 20-
times higher for Pyr (Table 3). High emissions of Pyr by heavy-duty vehicles have already been observed anywhere
else (Liacos et al., 2012; Cui et al., 2017). While the comparison between EFs determined during different
conditions is not straightforward in absence of gas phase measurements due to the high vapour pressure of the
lowest molecular weight organics (Fujitani et al., 2012; Polo Rehn, 2013), quite good agreements are found
between chassis dynamometer measurements and traffic-EFs; and with data from other studies. Indeed, again the
light-duty traffic-EFs for Fla and Pyr are quite consistent with the ones of the diesel Euro 3 measured with chassis
dynamometer. The light-duty traffic-EF for An is proportionally lower, somewhere between the ones of test Euro
3 and Euro 4 vehicles. The average EF for Pyr for the Euro 3 diesel vehicle of this study (Table 4) is close to the
ones of Perrone et al. (2014) for Euro 3 diesel vehicles (740±300 ng/km for passenger cars and 1810±570 ng/km



for commercial utility vehicles). Traffic EFs for Fla and Pyr are also in good agreement with average EFs in a
tunnel in China (40-50 km.h$^{-1}$/784-2776 veh.h$^{-1}$/18.4-35.1% HDV, He et al., 2008). These satisfying agreements
suggest that EFs determined in this research may be quite well representative of in-use vehicles. However, gas-
particle partitioning and photochemical degradation processes may make difficult the use of quantitative data for
these markers, as already observed (Zhang et al., 2005; Phuleria et al., 2007; Katsoyiannis et al., 2011; Tobiszewski
and Namieśnik, 2012).
In agreement with previous observations (Phuleria et al., 2006 and 2007; He et al., 2008; El Haddad et al., 2009;
Alves et al., 2016; Pant et al., 2017), 17α,21β-norhopane and 17α,21β-hopane are the two dominant hopanes
associated with traffic emissions. Very few traffic-EFs are available for hopanes in the recent literature and most
data available from recent studies are related to fuel consumption (e.g. Phuleria et al., 2006 and 2007). Traffic
hopane EFs of this study are much lower than those computed for tunnels in China (He et al. 2008; Cui et al. 2016).
In order to compare with European tunnel data, traffic-EFs are normalized by OC emissions. Value for the
17α,21β-norhopane (246 µg per g of OC) is similar to the one for a tunnel in Lisbon (Alves et al., 2016) but almost
half the ones for tunnels in Marseille and Birmingham (El Haddad et al., 2009; Pant et al., 2017). Conversely, the
value for 17α,21β-hopane (301 µg per g of OC) is very close to the ones found in tunnels in Birmingham and
Marseille but twice the one for the tunnel in Lisbon. These divergences and the quantification of heavy-duty traffic
emissions require further work in order to be able to define specific data for these indicators.
3.2.4 Other non-exhaust emissions
The average traffic-fleet EFs for $Ca^{2+}$ is the fifth highest traffic-fleet emission factor after EC, OC, Fe and $NO_3^-$.
While the emissions of $Ca^{2+}$ could not be discriminated between light-duty and heavy-duty traffics
(autocorrelation), this much larger emission factor than the ones measured in the exhaust of diesel vehicles is in
agreement with its origin. If $Ca^{2+}$ mainly comes from road dust resuspension, it does not only depend on the
intensity of traffic but also on the amounts of calcium deposited on the road (silt loading). Bukowiecki et al. (2009)
used the mass concentration measured 1-hour earlier in order to empirically account for the observed
autocorrelation behaviour in the calcium time series due to the accumulation of resuspended dust. The time series
resolution of 4 hours for collected particles in this study does not allow such calculations. Note that contrary to
Bukowiecki et al., most other trace elements that could be regressed in this study do not have the same behaviour
as $Ca^{2+}$. This could be explained by the removal of road dusts thanks to frequent rain events before and during the
sampling period.
Since EU tyres contain about 1% zinc oxide (Pant and Harrison, 2013) and Zn is the most abundant metallic
element in tyres commercialized in the U.S. (Apeagyei et al., 2011), Zn is often proposed as a key tracer of tyre
wear emissions. In this study, similarly to other works (Boogaard et al., 2011; Amato et al., 2011), Zn
concentrations did not show any roadside increment. This suggests the prevalence of other sources of Zn in the
Grenoble-Alpes conurbation, as well as in other urban environments. Further research are needed to determine
proper tracers of tyre wear emissions.
**4. Conclusions**
A large comprehensive particulate matter dataset has been collected from a simultaneous near-road and urban
background measurement field campaign, together with chassis dynamometer experiments. Near-road
measurements are made near a freeway with various driving conditions from free-flowing to stop-and-go traffic,



including frequent and severe braking events during periods of congestion (morning and afternoon commuting
times of workdays). This study attempted at determining emission factors for many chemical species from road
traffic and identifying and quantifying tracers of exhaust and non-exhaust vehicular emissions to be used in source
apportionment studies.
The traffic shows the larger emission factor for EC that is strongly associated with diesel traffic. The emission
factor for EC for the light-duty traffic is similar to the ones of passenger diesel cars non-retrofitted with particle
filter. The traffic-fleet EF for OC is slightly larger than those deduced from exhaust measurement of test vehicles.
This later observation would require further investigations in order to delineate the several possible causes for such
observation. In this environment dominated by the diesel traffic, the OC/EC ratio is below 0.4 (incremental
concentrations or emissions), as it is in diesel exhaust emissions. However, this ratio depends on the traffic fleet
and then may change in the future with the progressive introduction of vehicles retrofitted with particle filter.
Results showed the important contribution of metals from brake wear to particulate vehicular emissions. In
particular, Fe has the third higher traffic emission factor after EC and OC. Total brake wear emissions are estimated
for the RN87 highway, they are on average almost twice the particle emission standards for the exhausts of newer
vehicles (from Euro 5). We have shown that Cu is another important contributor to $PM_{10}$ from traffic and it could
be an excellent tracer of brake wear emissions in most environments. Even though they are less specific than Cu,
other metals such as Fe and Sn may be used to trace brake wear emissions through typical ratios. In particular,
ratios Cu/Fe of about 0.041 for light-duty traffic may possibly be the best option for estimating the brake wear
emissions due to the traffic of European passenger cars. Constant Cu/Fe ratios at different sites suggest similar
composition of brake elements at the European scale (as longer as Cu-free brakes are not spread in Europe). The
ratio Cu/Sn may possibly be another good option that requires further investigation. The use of both Cu/Mn and
Cu/Sb ratios seems to be less obvious since data are more scattered possibly due to frequent occurrences of other
influential source for Mn, and due to the introduction of Sb-free brake pads.
Recent literature show that some redox-active metals from brake wear (Cu, Fe, Mn, Cr) are associated with
oxidative stress (Poprac et al., 2017), high oxidative potential (Shirmohammadi et al., 2017; Weber et al., 2018),
and inflammation in lung tissue (Pardo et al., 2015). Also brake wear debris are associated with many health
outcomes (Kukutschová et al., 2009). Hence the impact of brake metals emissions on health requires a better
understanding, and the regulation of brake wear emissions needs to be considered.
Particulate organic emission data for European motor vehicles is scarce. In this study, the emission factors of a
few PAHs, n-alkanes and hopanes have been quantified. In agreement with previous works, low molecular weight
PAHs (mainly An, Fla, Pyr) are associated with diesel exhaust emissions. Pyrene and Fluoranthene are the ones
the most strongly associated with fresh diesel exhaust emissions. Pyrene is more largely emitted by the heavy-duty
traffic. Similarly to PAHs, n-alkanes from C19 to C26 are also associated with diesel vehicles, even though the
determination of the concentration of the dominant alkanes (from C20 to C23) strongly depends on measurement
conditions. While the comparison with other recent studies is difficult in the absence of gas quantification for *in*
*situ* measurements, a quite good agreement is found for most organics. However, the changing of the gas-particle
partitioning of low molecular PAHs and n-alkanes according to ambient temperature and dilution factors, and their
possible short-term photochemical degradation may compromise their use as quantitative markers of fuel
combustion.



Hopanes are markers of lubricating oil in the emissions of high-emitting vehicles (Rogge et al. 1993a; Zielinska
et al., 2004). In this study two hopanes ($17\alpha,21\beta$-norhopane and $17\alpha,21\beta$-hopane) have been clearly associated
with the traffic and more closely related to the heavy-duty traffic. However the quantification showed divergences
with other studies that require a better understanding.
Overall, this study delivered valuable information that could help in a better definition of traffic emissions in
source apportionment studies.
**Acknowledgements**
This work was funded by CORTEA-ADEME (PM-DRIVE program 1162C0002) that includes the funding of
chassis dynamometer and near-road field/urban background campaigns and PREDIT (MOCoPo programme) that
includes the near-road field measurements of regulated pollutants and traffic characteristics. Lucie Polo-Rehn's
PhD was funded by the Région Rhône-Alpes. Rain data was supplied by Météo France. The authors would like to
thank Patrick Tassel, Pascal Perret and Mathieu Goriaux (chassis dynamometer experiments), Julie Cozic and
Jean-Charles Francony (sample analyses) for their contribution to this work. We also thank Michel André for
supplying COPCETE emission factors. Part of the chemical analysis was performed on equipment provided by
Labex OSU@2020 (ANR10 LABX56).

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





# 2     **Tables and figures**

| Pollutant | $N_T$ | $N_{UB}$ | Median increment (%) | Sign test p value | Rank Wilcoxon test p value | R with total traffic | R with HDV | R with $NO_x$ | R with E | Remarks |
|---|---|---|---|---|---|---|---|---|---|---|
| OC | 57 | 57 | 22.7 | 0.000 | 0.000 | 0.00 | 0.16 | 0.37** | 0.38** | |
| EC | 57 | 57 | 67.8 | 0.000 | 0.000 | 0.50** | 0.89** | 0.89** | ----- | R(Cu/EC)=0.83** |
| $NO_x$ | 55 | 55 | 73.9 | 0.000 | 0.000 | 0.49** | 0.71** | ----- | 0.89** | |
| $PM_{10}$ | 55 | 55 | 18.7 | 0.000 | 0.000 | 0.33* | 0.33* | 0.55** | 0.65** | |
| $PM_{2.5}$ | 48 | 48 | 21.3 | 0.000 | 0.000 | 0.38** | 0.31* | 0.62** | 0.74** | |
| $PM_{coarse}$ | 48 | 48 | 8.2 | 0.041 | 0.115 | 0.27 | 0.29 | 0.26 | 0.24 | |
| $NO_3^-$ | 48 | 48 | 28.4 | 0.000 | 0.000 | 0.18 | 0.27 | 0.23 | 0.21 | |
| $Ca^{2+}$ | 57 | 26 | 34.2 | 0.011 | 0.005 | 0.42** | 0.63** | 0.56** | 0.44** | R(Ca/Cu)=0.64** |
| $Na^+$ | 57 | 26 | 33.2 | 0.001 | 0.003 | 0.14 | 0.30* | 0.19 | 0.04 | |
| $Mg^{2+}$ | 48 | 48 | 43.1 | 0.000 | 0.000 | 0.24 | 0.29* | 0.17 | 0.33* | $R(Mg^{2+}/NO_3^-)$=0.63**; $R(Mg^{2+}/Ti)$=0.57** |
| Ba | 57 | 26 | 37.2 | 0.035 | 0.013 | 0.49** | 0.26 | 0.25 | 0.35** | R(Ba/Cu)=0.57**; R(Ba/Sn)=0.74**; R(Ba/Fe)=0.71** |
| Co | 57 | 26 | 35.6 | 0.003 | 0.000 | 0.15 | 0.07 | 0.00 | 0.04 | $R(Co/SO_4^{2-})$=0.46**; R(Co/oxalates)=0.51**; $R(Co/NH_4^+)$=0.41** |
| Cr | 57 | 26 | 86.2$ | 0.000 | 0.000 | 0.03 | 0.25 | 0.31* | 0.40** | R(Cr/Mn=0.52**); R(Cr/Fe)=0.45** |
| Cu | 57 | 26 | 68.9 | 0.000 | 0.000 | 0.67** | 0.64** | 0.79** | 0.83** | R(Cu/Sn)=0.95**; R(Cu/Sb)=0.79**; R(Cu/Ti)=0.68**; R(Cu-An,Fla,Py, H3, H4)>0.5** |
| Fe | 57 | 26 | 84.1 | 0.000 | 0.000 | 0.68** | 0.60** | 0.67** | 0.74** | R(Cu/Fe)=0.86**; R(Fe/Sn)=0.92** |
| Mn | 57 | 26 | 59.6 | 0.000 | 0.000 | 0.48** | 0.66** | 0.78** | 0.77** | |
| Sb | 57 | 26 | 63.3 | 0.000 | 0.000 | 0.44** | 0.61** | 0.71** | 0.62** | R(Sb/Sn)=0.81** |
| Sn | 57 | 26 | 54.1 | 0.000 | 0.000 | 0.71** | 0.70** | 0.77** | 0.79** | |
| Sr | 57 | 26 | 89.2$ | 0.000 | 0.000 | 0.34** | 0.41** | 0.31* | 0.28* | |
| Ti | 57 | 26 | 60.8 | 0.003 | 0.001 | 0.58** | 0.59** | 0.44** | 0.45** | R(Ti/Al)=0.42** |

Table 1a: Median increments to concentrations measured near the RN87 highway in comparison to those measured in the urban background site (given in % of total concentrations) for species for which concentrations are significantly higher at the near-traffic site according to Sign and Rank Wilcoxon tests (p<0.05). $PM_{coarse}$ (difference between $PM_{10}$ and $PM_{2.5}$) is also included in this table. R are Spearman correlation coefficients with traffic, heavy-duty traffic (HDV), nitrogen oxides ($NO_x$), and elemental carbon (EC). $N_T$ : number of samples collected at the traffic site (Echirolles), $N_{UB}$ : number of samples collected at the urban background site (Les Frênes site). $: concentrations at Les Frênes are most of the time below detection limit. * mean statistically significant at the 95% level, ** at the 99% level.





| Pollutant | $N_T$ | $N_{UB}$ | Median increment (%) | Sign test p value | Rank Wilcoxon test p value | R with total traffic | R with HDV | R with $NO_x$ | R with EC | Remarks |
|---|---|---|---|---|---|---|---|---|---|---|
| Phe (phenanthrene) | 56 | 40 | 43.8 | 0.000 | 0.000 | 0.17 | 0.17 | 0.25 | 0.32* | |
| An (anthracene) | 56 | 40 | 55.0 | 0.000 | 0.000 | 0.34* | 0.39** | 0.48** | 0.51** | |
| Fla (fluoranthene) | 56 | 40 | 70.3 | 0.000 | 0.000 | 0.38** | 0.43** | 0.51** | 0.54** | |
| Pyr (pyrene) | 56 | 40 | 81.3 | 0.000 | 0.000 | 0.40** | 0.46** | 0.50** | 0.51** | |
| BaA (benzo(a)anthracene) | 56 | 40 | 55.3 | 0.000 | 0.000 | 0.16 | 0.23 | 0.29* | 0.39** | correlation with Levoglucosan : 0.75** |
| C17 (heptadecane) | 56 | 40 | 78.7 | 0.005 | 0.001 | -0.03 | 0.09 | 0.21 | 0.24 | |
| C18 (octadecane) | 56 | 40 | 39.6 | 0.010 | 0.004 | 0.14 | 0.22 | 0.36** | 0.43** | |
| C19 (nonadecane) | 56 | 40 | 83.1 | 0.000 | 0.000 | 0.23 | 0.29* | 0.39** | 0.39** | |
| C20 (icosane) | 56 | 40 | 70.0 | 0.000 | 0.000 | 0.24 | 0.39** | 0.50** | 0.42** | |
| C21 (henicosane) | 56 | 40 | 89.0 | 0.000 | 0.000 | 0.21 | 0.33* | 0.46** | 0.41** | |
| C22 (docosane) | 56 | 40 | 79.3 | 0.000 | 0.000 | 0.27* | 0.41** | 0.52** | 0.43** | |
| C23 (tricosane) | 56 | 40 | 66.6 | 0.000 | 0.000 | 0.31* | 0.40** | 0.61** | 0.59** | |
| C24 (tetracosane) | 56 | 40 | 68.3 | 0.000 | 0.000 | 0.26 | 0.35* | 0.50** | 0.63** | |
| C25 (pentacosane) | 56 | 40 | 48.2 | 0.000 | 0.000 | 0.24 | 0.29* | 0.55** | 0.61** | |
| C26 (hexacosane) | 56 | 40 | 70.3 | 0.000 | 0.000 | 0.22 | 0.36** | 0.62** | 0.63** | |
| C27 (heptacosane) | 56 | 40 | 36.4 | 0.000 | 0.000 | 0.17 | 0.18 | 0.36** | 0.44** | |
| C28 (octacosane) | 56 | 40 | 57.4 | 0.026 | 0.000 | -0.15 | -0.01 | 0.19 | 0.22 | correlation with Levoglucosan : 0.55** |
| C29 (nonacosane) | 56 | 40 | 32.0 | 0.006 | 0.000 | 0.15 | 0.16 | 0.24 | 0.26 | correlation with Levoglucosan : 0.39** |
| C30 (triacontane) | 56 | 40 | 54.7 | 0.002 | 0.000 | 0.03 | 0.16 | 0.33* | 0.38** | correlation with Levoglucosan : 0.51** |
| C31 (hentriacontane) | 56 | 40 | 20.4 | 0.007 | 0.003 | 0.14 | 0.21 | 0.35* | 0.41** | correlation with Levoglucosan : 0.48** |
| C32 (dotriacontane) | 56 | 40 | 63.4 | 0.004 | 0.002 | -0.18 | -0.06 | 0.13 | 0.21 | correlation with Levoglucosan : 0.47** |
| C33 (tritriacontane) | 56 | 40 | 34.4 | 0.030 | 0.008 | -0.03 | 0.07 | 0.23 | 0.32* | correlation with Levoglucosan : 0.47** |
| H3 (17α21βNorhopane) | 56 | 40 | 97.4$ | 0.000 | 0.000 | 0.29* | 0.46** | 0.67** | 0.67** | |
| H4 (17α21βhopane) | 56 | 40 | 97.8$ | 0.000 | 0.000 | 0.22 | 0.37** | 0.59** | 0.61** | |

Table 1b: Median increments to concentrations measured near the RN87 highway in comparison to the urban background site (given in % of total concentrations at the traffic site) for organic species for which concentrations are significantly higher at the traffic site according to Sign and Rank Wilcoxon tests (p<0.05). R are Spearman correlation coefficients with traffic, heavy-duty traffic (HDV), nitrogen oxides ($NO_x$), and elemental carbon (EC). $N_T$ : number of samples collected at the traffic site (Echirolles), $N_{UB}$ : number of samples collected at the urban background site (Les Frênes site). $: concentrations at Les Frênes are most of the time below detection limit. * means statistically significant at the 95% level, ** at the 99% level.





| EFs | OC | EC | $NO_3^-$ | $Na^+$ | $Mg^{2+}$ | $Ca^{2+}$ |
|---|---|---|---|---|---|---|
| median | 16.8 | 39.0 | 1.82 | 0.176 | 0.061 | 1.62 |
| IQR | 8.4-27.8 | 35.5-45.5 | 0.45-2.6 | 0.104-0.394 | 0.044-0.106 | 0.80-2.27 |

Table 2a: Median emission factors for OC, EC and major ions significantly related to traffic and Interquartile Range (IQR : 25[th] and 75[th] percentiles) in mg.veh$^{-1}$.km$^{-1}$

| EFs | Ba | Cr | Cu | Fe | Mn | Sb | Sn | Sr | Ti |
|---|---|---|---|---|---|---|---|---|---|
| median | 66 | 43 | 300 | 6711 | 62 | 27 | 55 | 11 | 28 |
| IQR | 20-136 | 38-75 | 228-450 | 4527-9302 | 51-72 | 19-39 | 40-81 | 8-16 | 22-40 |

Table 2b: Median emission factors for elements significantly related to traffic and Interquartile Range (IQR : 25[th] and 75[th] percentiles) in µg.veh$^{-1}$.km$^{-1}$

| EFs | Phe | An | Fla | Pyr | H3 | H4 |
|---|---|---|---|---|---|---|
| median | 0.716 | 0.054 | 1.69 | 2.54 | 3.46 | 4.34 |
| IQR | 0.346-1.15 | 0.033-0.085 | 1.14-2.37 | 1.78-3.28 | 1.99-6.90 | 2.67-6.89 |

Table 2c: Median emission factors for PAHs significantly related to traffic and 17α21βNorhopane (H3), 17α21βhopane (H4) and their Interquartile Range (IQR : 25[th] and 75[th] percentiles) in µg.veh$^{-1}$.km$^{-1}$

| EFs | C19 | C20 | C21 | C22 | C23 | C24 | C25 | C26 |
|---|---|---|---|---|---|---|---|---|
| median | 3.6 | 9.5 | 30.3 | 42.5 | 48.0 | 39.9 | 19.8 | 12.7 |
| IQR | 2.0-8.3 | 5.2-18.9 | 19.9-62.4 | 29.8-81.4 | 30.6-70.3 | 26.1-52.5 | 12.5-34.2 | 9.3-21.9 |

Table 2d: Median emission factors for n-alkanes significantly related to traffic and Interquartile Range (IQR : 25[th] and 75[th] percentiles) in µg.veh$^{-1}$.km$^{-1}$



| | Unit | $r^2$ | HDV | | | | | LDV | | | | |
|---|---|---|---|---|---|---|---|---|---|---|---|---|
| | | | Coeff. | SD | $p$ | conf.inter. 95% | | Coeff. | SD | $p$ | conf.inter. 95% | |
| EC | mg.veh$^{-1}$.km$^{-1}$ | 0.92 | 148.4 | 22.7 | 0.000 | 102.9 | 193.9 | 30.2 | 1.9 | 0.000 | 26.4 | 34.0 |
| Cu | µg.veh$^{-1}$.km$^{-1}$ | 0.73 | 3371 | 990 | 0.003 | 1312 | 5430 | 258 | 104 | 0.021 | 42 | 474 |
| Fe | mg.veh$^{-1}$.km$^{-1}$ | 0.81 | 48.0 | 19.1 | 0.023 | 7.5 | 88.5 | 6.3 | 1.7 | 0.002 | 2.7 | 9.8 |
| Sb | µg.veh$^{-1}$.km$^{-1}$ | 0.77 | 246 | 72 | 0.003 | 96 | 395 | 20.1 | 6.9 | 0.006 | 5.6 | 34.5 |
| Sn | µg.veh$^{-1}$.km$^{-1}$ | 0.82 | 512 | 149 | 0.003 | 198 | 825 | 54 | 15 | 0.002 | 23 | 85 |
| Pyr | µg.veh$^{-1}$.km$^{-1}$ | 0.60 | 28.4 | 6.3 | 0.000 | 15.6 | 41.1 | 1.459 | 0.658 | 0.034 | 0.121 | 2.798 |
| An | µg.veh$^{-1}$.km$^{-1}$ | 0.55 | 0.435 | 0.128 | 0.002 | 0.174 | 0.697 | 0.0352 | 0.0131 | 0.011 | 0.0086 | 0.0618 |
| Fla | µg.veh$^{-1}$.km$^{-1}$ | 0.58 | 7.38 | 3.38 | 0.037 | 0.49 | 14.27 | 1.254 | 0.305 | 0.000 | 0.632 | 1.875 |
| C23 | µg.veh$^{-1}$.km$^{-1}$ | 0.66 | 277.8 | 70.6 | 0.000 | 133.8 | 421.7 | 23.9 | 6.3 | 0.001 | 11.1 | 36.8 |
| C24 | µg.veh$^{-1}$.km$^{-1}$ | 0.62 | 163.9 | 54.0 | 0.005 | 53.7 | 274.1 | 19.2 | 4.8 | 0.000 | 9.3 | 29.0 |

Table 3: Results of the Multiple Linear Regressions with the heavy-duty traffic (HDV) and the light duty traffic (LDV): square correlation coefficients, unstandardized coefficients with standard deviations for HDV and LDV, p-values and confidence intervals at 95%.





| | Unit | E3D | | E4D | | E2P | | E4P | | E4D+PF | |
|---|---|---|---|---|---|---|---|---|---|---|---|
| | | median | IQR | median | IQR | median | IQR | median | IQR | median | IQR |
| EC | mg/km | 31.6 | 29.7-41.2 | 27.1 | 15.8-37.4 | <QL | <QL | 0.06 | 0-0.06 | 0.37 | 0.12-0.44 |
| OC | mg/km | 10.7 | 7.3-14.4 | 5.6 | 3.3-6.0 | 0.3 | 0.2-0.4 | <QL | <QL | 0.14 | 0.07-0.18 |
| OC/EC | - | 0.29 | 0.24-0.33 | 0.16 | 0.12-0.19 | und. | und. | und. | und. | 0.61 | 0.40-1.94 |
| Ba | µg/km | 25.0 | 17.7-48.9 | 27.3 | 13.6-30.3 | 3. 6 | 2.8-5.0 | - | - | 0 | 0-0.2 |
| Co | µg/km | <DL | <DL | <DL | <DL | <DL | <DL | - | - | <DL | <DL |
| Cr | µg/km | 0.92 | 0.46-5.7 | 2.7 | 1.3-2.9 | 0.91 | 0.75-1.1 | - | - | 0.16 | 0.08-0.27 |
| Cu | µg/km | 9.2 | 5.2-12.9 | 1.1 | 0.56-2.4 | 0.40 | 0.33-0.79 | - | - | 0.14 | 0.10-0.29 |
| Fe | µg/km | 84 | 48-213 | 51 | 25-98 | 22 | 20-27 | - | - | 0.93 | 0.47-2.5 |
| Mn | µg/km | 0.56 | 0.49-4.3 | 4.2 | 2.4-4.3 | 0.42 | 0.13-0.81 | - | - | 0.07 | 0.03-0.09 |
| Sb | µg/km | <DL | <DL | <DL | <DL | <DL | <DL | - | - | <DL | <DL |
| Sn | µg/km | 0.92 | 0.46-5.4 | 2.25 | 1.1-2.6 | 0.06 | 0-0.14 | - | - | 0.06 | 0.03-0.08 |
| Sr | µg/km | 0.67 | 0.48-1.0 | 0.68 | 0.60-1.5 | 0.15 | 0.10-0.31 | - | - | 0.02 | 0.01-0.04 |
| Ti | µg/km | 8.3 | 7.4-10.7 | 15.6 | 12.0-28.1 | 2.8 | 1.5-4.4 | - | - | <DL | 0-0.1 |
| Ca$^{2+}$ | µg/km | <DL | 0-109 | 39 | 0-52 | <DL | <DL | <DL | <DL | 1.2 | 1.0-2.8 |
| Na$^+$ | µg/km | <DL | 0-89 | <DL | 0-60 | 3.6 | 2.7-6.0 | <DL | <DL | 1.5 | 0.34-6.8 |
| NO$_3^-$ | µg/km | 226 | 113-267 | 56.5 | 32-352 | 1.7 | 1.3-2.3 | <DL | <DL | 35 | 26-49 |
| Phe | µg/km | 3.53 | 3.36-4.33 | 0.125 | 0-0.356 | <QL | 0-0.005 | <QL | <QL | <QL | 0-0.001 |
| An | µg/km | 0.091 | 0.07-0.11 | 0.013 | 0.01-0.02 | <QL | <QL | <QL | <QL | 0.015 | 0-0.015 |
| Fla | µg/km | 0.956 | 0.37-1.27 | 0.072 | 0.05-0.11 | <QL | <QL | 0.001 | 0-0.002 | <QL | <QL |
| Pyr | µg/km | 1.066 | 0.26-1.3 | 0.110 | 0.03-0.19 | <QL | <QL | 0.003 | 0-0.003 | <QL | 0-0.001 |
| C19 | µg/km | 17.4 | 11.5-22.3 | 4.66 | 3.05-5.66 | 0.15 | 0-0.16 | <QL | <QL | 0.92 | 0-2.20 |
| C20 | µg/km | 25.4 | 14.3-29.7 | 5.10 | 3.36-6.36 | <QL | 0-0.27 | <QL | <QL | 2.78 | 0.36-3.25 |
| C21 | µg/km | 31.5 | 17.1-35.1 | 5.84 | 3.65-7.65 | <QL | 0-1.08 | <QL | <QL | 2.76 | 1.22-4.50 |
| C22 | µg/km | 22.1 | 13.0-25.8 | 4.94 | 2.90-6.39 | <QL | <QL | <QL | <QL | 2.20 | 0-2.44 |
| C23 | µg/km | 16.8 | 9.43-21.3 | 4.18 | 2.23-5.86 | <QL | 0-0.85 | <QL | 0-0.04 | 1.11 | 0-1.82 |
| C24 | µg/km | 11.1 | 6.19-15.1 | 3.51 | 1.70-4.69 | 0.13 | 0-0.31 | <QL | <QL | 0.22 | 0-0.70 |
| C25 | µg/km | 6.12 | 3.00-8.39 | 1.94 | 1.02-2.87 | 0.07 | 0-0.53 | 0.02 | 0.01-0.02 | <QL | <QL |
| C26 | µg/km | 3.57 | 1.91-3.85 | 1.23 | 0.76-1.56 | <QL | 0-0.05 | <QL | 0-0.01 | <QL | <QL |

Table 4: Average emissions factors (median and IQR: interquartile range) determined by chassis dynamometer measurements for different types of passenger cars: Euro 3 diesel (E3D), Euro 4 diesel (E4D), Euro 2 Petrol (E2P), Euro 4 Petrol (E4P) and Euro 4 diesel equipped with particle filter (E4D+PF) using urban, urban cold, road real-driving cycles. Und.: undetermined.





| Ratios on | Cu/Sb | Cu/Fe | Cu/Sn | Cu/Mn | OC/EC |
|---|---|---|---|---|---|
| Roadside concentrations | 10.4 ±4.5 | 0.045 ±0.015 | 4.7 ±1.0 | 3.6 ±1.6 | 1.00 ±0.49 |
| Incremental concentrations | 11.7 ±5.1 | 0.043 ±0.015 | 5.4 ±1.8 | 6.4 ±5.9 | 0.33 ±0.22 |
| LDV Emission factors | 12.8 ±5.2[$] | 0.041 ±0.010[$] | 4.8 ±1.1[$] | und. | und. |
| HDV Emission factors | 13.7 ±4.3[$] | 0.070 ±0.017[$] | 6.6 ±1.1[$] | und. | und. |
| Traffic emission factors | 12.6 ±4.7 | 0.046 ±0.015 | 5.6 ±1.8 | 5.7 ±2.9 | 0.44 ±0.32 |

Table 5: Ratios for road-side, incremental (road-side minus urban background) concentrations, and RN87-traffic emission factors, with standard deviations. [$]: using Taylor expansion for approx.: $var\left(\frac{x}{y}\right) = \frac{1}{\mu_y^2} var(x) + \frac{\mu_x^2}{\mu_y^4} var(y) - 2\frac{\mu_x}{\mu_y^3} Cov(x,y)$





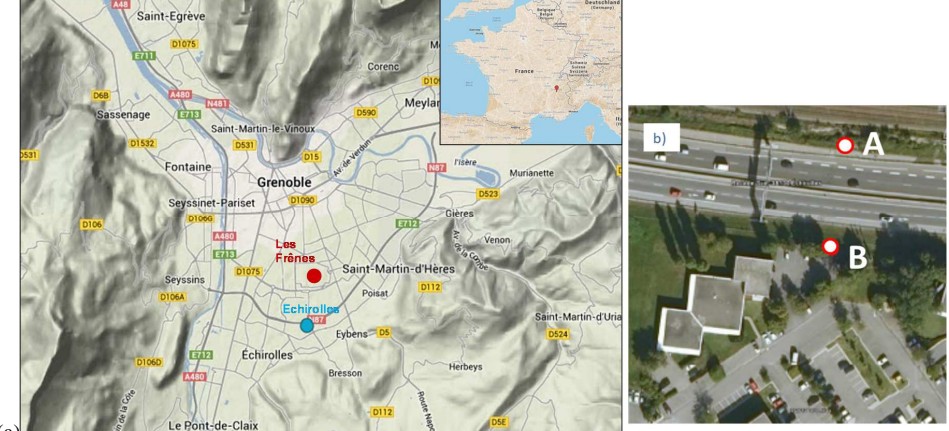

(a)

Figure 1: (a) Location of sampling sites in Grenoble-Alpes conurbation: the roadside site (Echirolles) and the urban background site (Les Frênes). (b) Echirolles sampling site: sampling and measurement devices are around the B point; the traffic electromagnetic loop monitoring are beneath the road (on the AB line), on the left, the cameras linked to automatic license plate recognition code are located on overhead gantries.



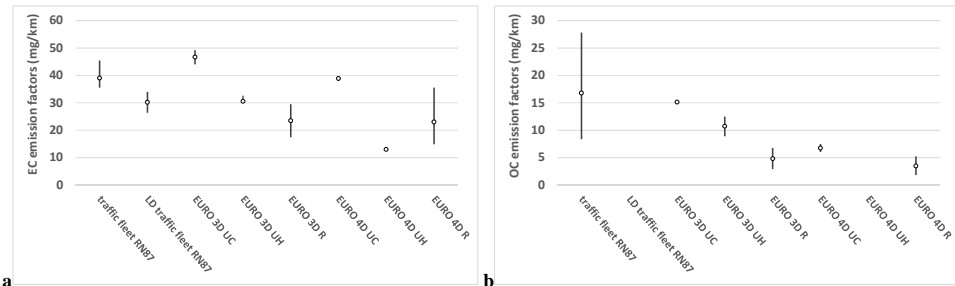

a                                                                     b

Figure 2: Emission factors for EC (a) and OC (b) for the mixed traffic fleet and the light duty traffic of the RN87/E712 freeway and from
5   the chassis dynamometer measurements of exhaust emissions of EURO 3 and EURO 4 vehicles for respectively urban cold (UC), urban hot
(UH) and road (R) driving cycles.

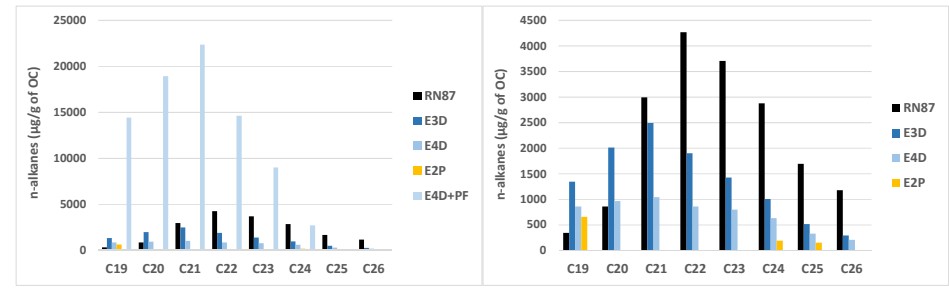

Figure 3: n-alkanes emissions normalized by OC emissions at the traffic site (RN87/E712) and from the exhaust of a Diesel Euro 3 vehicle
(E3D), a Diesel Euro 4 vehicle (E4D), a Diesel Euro 4 vehicle equipped with a particle filter (E4D+PF) (on the left only) and a Euro 2 Petrol
vehicles.

