# Peer review of "Identification and quantification of particulate tracers of"

_Atmospheric Chemistry and Physics, 2018_

## Referee Comment (RC1) · Anonymous Referee #2 · 30 Oct 2018

–GENERAL COMMENTS– Charron et al present chemical speciation of particulate matter collected from chassis dynamometer experiments as well as near-road and urban background sites. Such measurements are important for understanding both exhaust and non-exhaust emissions and their variability across changing and variable vehicle fleets as well as locations with different traffic conditions and meteorology. The wide range of measured species (which are effectively contextualized within other measurements in the literature) strengthens the dataset, but also presents challenges for clear communication. A number of editorial comments are given in order to clarify ambiguous or unclear meaning as well as to correct grammatical errors. Improving the clarity of figures would also strengthen the manuscript.

[Figure]

–SPECIFIC COMMENTS–

SECTION 2.1.1 and SECTION 3.2.3: (1) Please provide the sample flowrates, dilution ratios, and sampling time duration for each vehicle tested (perhaps in supplemental section I). (2) The authors note that differences in dilution ratios could affect the distribution of n-alkanes due to differences in phase partitioning among species with different vapor pressures. Differences in filter face velocity could also lead to differences in vapor adsorption (positive artifacts) and particle evaporation (negative artifacts), resulting in different degrees of over-estimation of particle-phase organic material across different sampling conditions (and thus different vehicle types). For chassis dynamometer experiments, filter face velocity differs by up to a factor of 1.67 for molecularly speciated measurements and up to a factor of 6 for EC and OC measurements. Would quartz filter sampling artifacts due to differences in sample flowrates (ie. face velocity, pressure drop) affect the authors' conclusions about distribution of n-alkanes as well as OC/EC values? (In particular, how would quartz filter artifacts affect the authors' conclusions about the impact of the particulate filter retrofit on OC and EC emissions?) REFERENCE Solomon, P. A., et al., Evaluation of PM2.5 chemical speciation samplers for use in the U.S. EPA national PM2.5 chemical speciation network, EPA Rep. EPA-454/R-01-005, Off. of Air Qual. Plann. and Stand., Research Triangle Park, N. C., 2000. REFERENCE McDow, S. R., and J. J. Huntzicker, Vapor adsorption artifact in the sampling of organic aerosol: face velocity effects, Atmos. Environ., Part A, 24, 2563 – 2571, 1990.

PAGE 5, LINE 9: Please specify which analysis method (GCMS or LCMS) is used for which organic molecules.

SECTION 2.4: Which type of MLR analysis was used in this work? (ie. Which algorithm was used to calculate the MLR relationships with HD and LD traffic?)

PAGE 9, LINE 17-18: Which constants have high p values? Those in Table 3 all have low (significant) values. In addition, the second half of the sentence implies that the

authors are comparing the urban background and remote site? Please clarify.

PAGE 10, LINE 5: What do the authors mean by "smoker vehicles"?

PAGE 11 LINES 25-29: 1) Which "unquantified compounds"? How is this rough estimate calculated if these compounds are not quantified? 2) I suggest also including the measured particulate EF's for exhaust and non-exhaust emissions to compare both exhaust and non-exhaust EF's with the standard.

PAGE 16 LINES 3-4: Please be more specific about which divergences were observed.

–COMMENTS ON TABLES AND FIGURES–

TABLE 1A: 1) 10th column's title should be "R with EC". 2) Why are parts of the table highlighted? Please include this in the caption.

TABLE 1B: Why are parts of the table highlighted? Please include this in the caption.

TABLE 3: Perhaps this would be addressed in the type-setting process, but please ensure that the units column is sufficiently wide, center the column titles, and use heavier borders to separate the HDV and LDV sections.

TABLE III-1: The title should be for a figure, not a table. Also, please indicate both data series in the legend (currently only harmonic mean speed is included).

FIGURE 2: Please indicate in the figure or caption why there are missing data points in Figure 2B (or leave these categories out of the plot). Suggestions to increase readability: Consider combining EC and OC into one plot. Instead of repeating vehicle type for each set of three conditions (UC, UH, R), consider grouping them with a bracket and labeling them together with the vehicle type.

SUPPLEMENTAL VII-1 and VII-2: Please use different colors for different species (ie. use a single color for Cu and not for any other species). Please also increase the text size of axes titles and tick mark labels.

SUPPLEMENTAL VIII: Please increase the text size of axes titles, tick mark labels, and other text.

SUPPLEMENTAL IX: Given the attention paid to ratios in this study, I would suggest adding rows for the most relevant ratios (at the least, Cu/Fn, Cu/Mn, and Cu/Sn) where data is available.

–EDITORIAL COMMENTS–

PAGE 1, LINE 13: grammar: add comma: "vehicular emissions, a large comprehensive dataset"

PAGE 1, LINE 20: ambiguous, replace "Most of the first ones" with "Light-duty traffic emission factors" and move "in absence of significant non-combustion emissions" to be beginning of the sentence (since it applies to the traffic emission factors, not the chassis dynamometer measurements)

PAGE 1, LINE 21: suggested change to correct grammar and increase clarity: "Since recent measurements in Europe including those from this study are consistent, ratios involving copper (Cu/Fe and Cu/Sn) could be used as brake-wear emissions tracers as long as brakes with Cu remain in use."

PAGE 1, LINE 23: The sentence regarding OC/EC ratio does not seem relevant or necessary to the abstract. In addition, the language implies that the OC/EC ratio is always 0.44 in France, which is likely not the intention of the authors.

PAGE 1, LINE 26: grammar: change "markers; while, their" to "markers, since"

PAGE 1, LINE 28: grammar: "environments" (should be plural as written)

PAGE 1, LINE 30: grammar: "filters have been progressively introduced"

PAGE 1, LINE 36: soften tone: delete "It is obvious that"

PAGE 2, LINE 1: grammar: "Also, knowledge of the deleterious impacts of PM on

human health"

PAGE 2, LINE 2: grammar: "PM is responsible"

PAGE 2, LINE 11: grammar: "They also do not represent the variability"

PAGE 2, LINE 19: clarity / word choice: replace "chemistry of PM" with "chemical composition of exhaust and non-exhaust particulate emissions"

PAGE 4, LINE 26: clarity: replace "below installation of" with "below this threshold for"

PAGE 4, LINE 28: grammar: "30% and 36% respectively"

PAGE 4, LINE 33: "(see SI section IV)"

PAGE 5, LINE 12: grammar: replace "sampler" with "samples"

PAGE 5, LINE 13: ambiguity, makes it sound like there are two background and two urban sites: delete "two"

PAGE 6, LINE 11: I am not sure if perhaps the authors intended "residual" instead of "residues"?

PAGE 6, LINE 33: Is the discussion of the additional Fe and Mn source in the subsequent paragraph? I do not see it in Section 3.2 as indicated in the text.

PAGE 7, LINE 1: clarity / ambiguity: "are more scattered possibly due to differences between light-duty and heavy-duty emissions factors"

PAGE 7, LINE 12: precision / clarity: replace "relationships" with "correlations"

PAGE 7, LINES 33-34: Do the authors intend that the n-alkanes and hopanes are correlated to each (traffic, NOx, EC)? If so, change to "NOx, and EC".

PAGE 8, LINE 25: grammar: concentrations should be singular: "one for Sr concentration"

PAGE 8, LINE 36: grammar: "All of these suggest"

PAGE 9, LINES 8-9: To increase clarity, consider moving this sentence to the end of the first paragraph of this subsection (3.1.2).

PAGE 9, LINE 18: grammar: "contributes"

PAGE 9, LINE 27: change "technics" to "techniques" for more common spelling

PAGE 9, LINE 37: grammar: "larger than could be expected"

PAGE 10, LINE 2: grammar: "as ours, and the EF for exhaust OC"

PAGE 10, LINE 5: grammar: "contribution of smoker vehicles; and rapid formation"

PAGE 10 LINE 11-12: grammar: non-retrofitted is less clear than "without" (see comment on PAGE 15 LINE 6 as well): "test diesel vehicles without particle filters"

PAGE 10 LINE 26: grammar: "third highest traffic emission rate"

PAGE 11 LINE 28: grammar: "EF"

PAGE 11, LINE 30 – PAGE 12, LINE 8: The second sentence of this paragraph (regarding Cu and Sb brake wear emissions) confused me because it suggested that the Cu/Sb ratio would be a good candidate as a tracer, which is not the case (as communicated later in the paragraph). I suggest being more direct with the conclusion earlier in the paragraph to avoid confusion.

PAGE 11 LINE 35: tone: suggestion to avoid the word "obviously." Also, please qualify the sentence by adding "in this study."

PAGE 11 LINE 31: Add citation for CITEPA in references

PAGE 11 LINE 35: grammar: "depend"

PAGE 12 LINE 6: replace "Then" with "Thus"

PAGE 12 LINE 23: replace "spent" with "used"

PAGE 12 LINE 31: precision / clarity: "Cu/Sn would be the strongest candidate

PAGE 14 LINE 12: grammar / clarity: "The normalized abundance of $17\alpha,21\beta$-norhopane (246 $\mu$g per g of OC) ..."

PAGE 14 LINE 17: replace "data" with "emissions factors and compositions"

PAGE 15 LINE 2: grammar: "This study determined ...identified ...quantified" or "This study attempted to determine...identify...quantify"

PAGE 15 LINE 6: see comment on PAGE 10 LINES 11-12 for similar issue: "passenger diesel cars without particle filters"

PAGE 15, LINE 5: I am not sure what the authors are communicating with this first sentence ("The traffic shows the larger emission factor..."). Please rephrase.

PAGE 15 LINE 19: As written, implies that different sites in Europe were sampled in this study. Instead: "Cu/Fe ratios consistent with literature values from other sites suggest similar brake composition for these elements throughout Europe (as long as Cu-free brakes do not increase in use)."

PAGE 15 LINES 21-23: Ambiguous as written as to whether these ratios are good or bad tracers. Instead: "Our measurements do not support the use of Cu/Mn and Cu/Sb as tracers of brake wear emissions possibly due to additional sources of Mn as well as the introduction of Sb-free brake pads."

PAGE 15 LINE 36: clarity: "agreement between chassis dynamometer and near-road measurements"

PAGE 15 LINE 36: delete "the change of the"

PAGE 16 LINES 5-6: I suggest a more specific concluding sentence. Perhaps replace "delivered valuable information" with "describes exhaust and non-exhaust emissions measurements"

---

## Referee Comment (RC2) · Anonymous Referee #1 · 3 Dec 2018

The study by Charron et al. documents measurements of PM10 samples collected at a roadside in France. Overall, the results appear to be of high quality and are of potential use to further source apportionment work. The authors contextualize the study with the increasing importance of non-exhaust PM to total vehicular emissions, which is an interesting and intuitive concept. Justification for sampling PM10 should be included in the paper. What is the typical size distribution for vehicular emissions? Exhaust emissions, especially those treated with a particulate filter onboard the vehicle are dominated by ultrafine particle emissions, which have vanishingly small mass, especially in comparison to the remainder of PM10. Therefore, does PM10 over-sample the resuspended and non-exhaust emissions? Are aerosol size distributions of non-

exhaust emissions known? The size-segregated composition of the PM in this study probably varies strongly, and will have an influence on the findings. Please discuss this aspect of the study in greater detail. Coarse particles also have fast deposition rates, limiting their influence on respiratory exposure compared to PM2.5 (or size classes with smaller upper size limits).

Inclusion of more data visualizations throughout the manuscript, especially related the early sections of results (temporal profiles) is strongly advised. The reviewer's comprehension of the text was heavily improved by opening the SI, which should not be a requirement of reading a paper. This recommendation will likely improve the manuscript by an important margin. Combining metals onto fewer panels may help so that the entirety of SI Section VII does not need to be included as-is (for instance). Please consider this idea for all sections of the manuscript.

The presentation of the manuscript overall could use some English grammar editing and overall proofreading. A number of small grammatical or usage errors exist throughout the document. A few salient ones are pointed out in the minor comments below, but this should not be considered an exhaustive list of corrections.

Overall opinion: This study should be considered for publication after addressing the comments of this review in a major revision.

Introduction: The study is contextualized well, focused on the importance of non-exhaust emissions from vehicles, which may now be the most important aspect of vehicular emissions. Findings were, however, associated with both exhaust and non-exhaust emissions, which is clear.

Methods: Methods were clearly described, with the exception of a lack of definition for TEOM-FDMS (a minor comment).

Results: The use of the term 'incremental' is confusing (occurs throughout 'Results'). Referring to the "increment" with a clearer name would greatly clarify the presentation

this important quantity.

Section 3.1.1: It would be helpful to see a graphical representation of the contribution of each component of PM10 to the total mass.

Page 6, Line 32: It is becoming clear that the influence of particle size may be evident in the data (also see broader comments above). Please include a description or reference to what is known about the size-resolved composition of vehicle emissions from both exhaust and non-exhaust sources. [Such a discussion is not necessarily relevant to insert at this point in the manuscript, however, its importance began to become clear at this point.]

Page 7, line 38 – Page 7, line 2: Can a consideration for super-emitting vehicles be included in this part of the discussion?

Page 8, lines 20-22: The measured 'dominant regional contribution' to OC may be masking the vehicular primary OC due to differences in the size distribution. Regional, secondary OC may be significantly aged, and therefore larger in size. While this will manifest as an overwhelming signal in a PM10 sample, it may not be so in a PM1 sample, or even a PM0.1 sample. Please discuss the significance of what is known about size-dependent composition of particles associated with vehicle emissions.

Page 8, lines 33-38: If Co does not show any correlation with traffic emissions, how can it be reasonably concluded from the data collected in the present study that Co is a contributor to brake wear? The authors clearly attempt to make a case using the literature. If the contribution of other elements to one another is going to used to explain their similar vehicular sources (Fe, Cu, etc), then the same standard must be held to Co. At best, perhaps there is some other, more consistent, low level source of Co that is flattening the temporal profile. A conclusion as written in this passage, however, is dubious.

Page 9, lines 11-15: Can the impact of variability within the parts of the vehicle fleet

be incorporated? In such highly controlled emissions from vehicles, a small number of super-emitters may be quite impactful. Page 9, lines 34-35: This is a salient point for air quality management and regulation.

Page 10, line 5 (and other instances): Please give a more succinct definition to the term "smoker vehicles". Perhaps the authors could call these super-emitters? (see also comment about Page 9, lines 11-15)

Page 11, line 22: The authors use and cite a finding that a dominant fraction of brake wear emissions come from the disc and not the brake pad. This seems hard to believe considering the fact that the brake pad is the primarily consumable part, and that brake discs (rotors) do not need to be replaced as often.

Page 11, line 35: Please clarify and/or define "PM fraction". This term is not used routinely in this manuscript. A change in wording may help in this instance.

Page 11, line 34 - Page 12, line 8: It may help to define Cu/Sb in the brake materials themselves. How might consistency in the brake materials themselves drive atmospheric Cu/Sb? Could the act of particle formation (temperature, breaking force, etc) influence the ratio?

Page 15, line 17-19: Has Cu/Fe been reported in any other proportion in the atmosphere? Please illustrate (perhaps in the results section on this topic) that the ∼4% value is unique to vehicles.

Page 15, line 24-28: This is a strange placement for an overview paragraph about the significance of redox-active metals, which have only been mentioned as such in the introduction. This paragraph is probably better off at the beginning of the conclusions section, mirroring the structure of the paper itself.

Page 15, line 29: While studies such as this one may be scarce in the literature, the authors have reported agreement with these studies throughout the manuscript – suggesting that the science is highly convergent. Please provide a clear, summary assessment of the novel findings of this study.

Page 16, lines 5-6: This concluding statement seems to highlight the fact that this is a characterization study with little in the way of entirely new findings. (see previous comment) Do the authors believe feel that this is the case? If not, a revised summary statement or declaration of a way forward in light of the present study is in order.

Minor comments: Line 20-21: "Most of the first ones" – please be specific, most of the first 'what'? What do you mean by first? I honestly do not know to which prior items this sentence refers.

Line 24-26: "On the contrary,..." to what??

Page 8, line 9: should refer to "SI Section VII"

Page 9, line 27: change "technics" to "techniques"

Page 10, line 2: define DPF, first usage

Page 10, line 26: change to "third highest"

---

## Author Comment (AC1) · 23 Jan 2019

–GENERAL COMMENTS– Charron et al present chemical speciation of particulate matter collected from chassis dynamometer experiments as well as near-road and urban background sites. Such measurements are important for understanding both exhaust and non-exhaust emissions and their variability across changing and variable vehicle fleets as well as locations with different traffic conditions and meteorology. The wide range of measured species (which are effectively contextualized within other measurements in the literature) strengthens the dataset, but also presents challenges for clear communication. A number of editorial comments are given in order to clarify

ambiguous or unclear meaning as well as to correct grammatical errors. Improving the clarity of figures would also strengthen the manuscript. RESPONSE: I thank the referee on behalf of all co-authors for the proposed very accurate amendments and the time spent on this paper.

–SPECIFIC COMMENTS– SECTION 2.1.1 and SECTION 3.2.3: (1) Please provide the sample flowrates, dilution ratios, and sampling time duration for each vehicle tested (perhaps in supplemental section I). RESPONSE: A table is added in the SI I (Table I-2).

(2) The authors note that differences in dilution ratios could affect the distribution of n-alkanes due to differences in phase partitioning among species with different vapor pressures. Differences in filter face velocity could also lead to differences in vapour adsorption (positive artifacts) and particle evaporation (negative artifacts), resulting in different degrees of over-estimation of particle-phase organic material across different sampling conditions (and thus different vehicle types). For chassis dynamometer experiments, filter face velocity differs by up to a factor of 1.67 for molecularly speciated measurements and up to a factor of 6 for EC and OC measurements. Would quartz filter sampling artifacts due to differences in sample flowrates (ie. face velocity, pressure drop) affect the authors' conclusions about distribution of n-alkanes as well as OC/EC values? (In particular, how would quartz filter artifacts affect the authors' conclusions about the impact of the particulate filter retrofit on OC and EC emissions?) REFERENCE Solomon, P. A., et al., Evaluation of PM2.5 chemical speciation samplers for use in the U.S. EPA national PM2.5 chemical speciation network, EPA Rep. EPA-454/R-01-005, Off. of Air Qual. Plann. and Stand., Research Triangle Park, N. C., 2000. REFERENCE McDow, S. R., and J. J. Huntzicker, Vapor adsorption artifact in the sampling of organic aerosol: face velocity effects, Atmos. Environ., Part A, 24, 2563 – 2571, 1990. RESPONSE: This is an interesting question from the reviewer. Quality sampling of exhaust PM is not straightforward and we attempted to make the best choices in order to have repeatable analyses of PM collected from the CVS of
the chassis dynamometer while considering the constraints associated with such sampling conditions. Filter face velocities were similar to, or not too far from, the one of atmospheric near-road measurements (47 cm/s) for chassis dynamometer samplings of 50 l/min (48 cm/s) and 40 l/min (38 cm/s). Then we can expect that this parameter will not significantly modify the partition of n-alkanes in comparison to near-road measurements since PM samplings for organic speciation are made at 40 and 50 l/min. Conversely, very low sample flow rates (5 and 10 l/min, corresponding to filter velocities of about 5 and 10 cm/s respectively) are used for PM samplings dedicated to EC and OC measurements in the exhaust of diesel vehicles non-retrofitted with PF (Euro 3 and Euro 4 diesel vehicles). Indeed, these very low filter velocities may possibly be responsible for enhanced adsorption of organic vapours by the quartz filters, leading to positive artefact for OC (McDow and Huntzicker, 1990; Turpin et al., 2000; Vecchi et al., 2009), while EC would not be affected by filter face velocity (Vecchi et al., 2009). However, the magnitude of such potential artefact is difficult to estimate (Viana et al., 2006) and especially for such conditions that are very different from those in the atmosphere (in particular the very high concentrations of organic gases and OC that are also likely to influence the adsorption equilibrium of the filter with the incoming gas-phase concentrations). If we admit that the positive artefact is more important for these samples, this means that the real OC/EC ratios would be even lower for non-retrofitted diesel vehicles. Therefore, this does not change the conclusion regarding the impact of PF equipment on OC and EC emissions. Therefore, we included the following comment in part 2.1.2 (Exhaust sampling), lines 20-27 (new version of manuscript): "Conditions of vehicle tests are detailed in the supplemental section (Table I-2). Filter face velocities for chassis dynamometer samplings dedicated to organic speciation were similar or close to the one of atmospheric near-road measurements, while, because of very high concentration levels, lower sample flow rates are used for PM samplings dedicated to EC and OC measurements in the exhaust of diesel vehicles non-retrofitted with PF (Euro 3 and Euro 4 diesel vehicles). These very low filter velocities may influence the adsorption of organic vapours by the Quartz filter (McDow and Huntzicker, 1990; Turpin

et al., 2000 ; Vecchi et al., 2009), while EC would not be affected by filter face velocity (Vecchi et al., 2009)." The new references are added.

PAGE 5, LINE 9: Please specify which analysis method (GCMS or LCMS) is used for which organic molecules. RESPONSE: The information is added page 5 lines 21-22. "The chemical speciation of organic particles are performed by Gas Chromatography–Mass Spectrometry (GC-MS), except PAHs that were measured by liquid chromatography (HPLC) using a fluorescence detector"

SECTION2.4: Which type of MLR analysis was used in this work? (ie. Which algorithm was used to calculate the MLR relationships with HD and LD traffic?) RESPONSE: A standard least square regression method is used since the influence of the two independent variables (light duty and heavy-duty traffic) was expected and in this case, no exploratory procedure (e.g. hierarchical or setwise regressions) was necessary. The sentence is modified as follows: (page 6, line 20-21)" Standard Multiple Linear Regression analyses (SPSS software) are performed…."

PAGE 9, LINE 17-18: Which constants have high p values? Those in Table 3 all have low (significant) values. In addition, the second half of the sentence implies that the authors are comparing the urban background and remote site? Please clarify. RESPONSE: The constants of all regressions have high p-values. As indicated (caption and in table 3), Table 3 does not present any constants (weak interest since they are all not significant), but only the coefficients of the regressions. All coefficients are significant. The authors compare the near-traffic site and the urban background site. In order to be better understood, I propose the following modification: I have transferred the sentence page 9 line 17-18 in the main part that presents the method (2.4 data analysis part) and I modified the text of this part. "The coefficients of the regressions represent average EFs for local heavy-duty and light duty traffics (Table 3). The constants represent the parts not related to local traffic. They are all not significant (p-values above 0.4 for metals and organics, p-value of 0.061 for EC) confirming the above assumption that mostly local traffic contributes to local increments in concentration."

PAGE 10, LINE 5: What do the authors mean by "smoker vehicles"? RESPONSE: Now "High emitting vehicles" replaces "smoker vehicles". (now line 15)

PAGE 11 LINES 25-29: 1) Which "unquantified compounds"? How is this rough estimate calculated if these compounds are not quantified? 2) I suggest also including the measured particulate EF's for exhaust and non-exhaust emissions to compare both exhaust and non-exhaust EF's with the standard. RESPONSE: 1) As already indicated in the text, the average brake profile data of Hulskotte et al. (2014) are used to estimate the average emission factor for brake wear assuming that the total proportion of metals is kept, and that 70% of the wear arise from the disc (as suggested by their research). The sentence with "unquantified compounds" is now replaced by (page 11- 12 lines 38-1): "So by adding to the sum of traffic-fleet EFs for metals related to brake wear the portion that corresponds to the elements and compounds not quantified in this study (C, S, Zn, Al, Si, Zr, Mo, V, Ni, Bi, W, P, Pb, Co), the rough estimation of 9.2 mg/km for emissions related to brake wear is got for the RN87 highway traffic." 2) The emission standards shown in the text correspond to the most recent vehicles (only 7% of Euro 5 in the traffic fleet during the campaign and virtually no Euro 6). This comparison is to show that in the near future, the contribution of non-exhaust emissions would have taken over the one at the exhaust. I propose this modification: "...the particle emission standards for the exhausts of the most recent vehicles (Euro 5 and Euro 6 vehicles, 5 mg/km)." The measured particulate EF's for exhaust are presented in Table 4 and traffic-fleet EFs (exhaust + non-exhaust) are presented in tables 2 and 3.

PAGE 16 LINES 3-4: Please be more specific about which divergences were observed. RESPONSE: Now it is specified in the text that the divergence concerns the ratios hopanes to OC (line 18-19). "However the quantification of ratios hopanes to OC showed divergences with other studies that require a better understanding."

–COMMENTS ON TABLES AND FIGURES– TABLE 1A: 1) 10th column's title should be "R with EC". 2) Why are parts of the table highlighted? Please include this in the caption. TABLE 1B: Why are parts of the table highlighted? Please include this in

the caption. RESPONSE: 1) that is right, thank you. 2) and Table 1b) These parts correspond to species significantly correlated with traffic indicators and for which local traffic contributions are above 50%. It was indicated page 7 lines 1-4. I propose in the new version of manuscript to include it in the caption and remove it from the text. In the captions: "Species significantly correlated with traffic indicators and for which local traffic contributions are above 50% are highlighted."

TABLE 3: Perhaps this would be addressed in the type-setting process, but please ensure that the units column is sufiňĄciently wide, center the column titles, and use heavier borders to separate the HDV and LDV sections. RESPONSE: Table 3 is now improved.

TABLE III-1: The title should be for a figure, not a table. Also, please indicate both data series in the legend (currently only harmonic mean speed is included). RE-SPONSE: That is right, it is corrected. Only harmonic mean speed is included since it is what best represent the traffic speed. Individual vehicles have different speeds, the magnitude of which depends on the traffic flow.

FIGURE2: Please indicate in the figure or caption why there are missing data points in Figure 2B (or leave these categories out of the plot). Suggestions to increase read-ability: Consider combining EC and OC into one plot. Instead of repeating vehicle type for each set of three conditions (UC, UH, R), consider grouping them with a bracket and labeling them together with the vehicle type. RESPONSE: The reasons why there are missing data points in Figure 2b are now indicated in the caption as follows: "The emission factor for OC for the LD traffic fleet could not be determined and no data available for the hot driving conditions for the Euro 4 diesel vehicles." Combining EC and OC into one plot would not increase readability, I prefer to keep two plots. The objective is to show how the traffic emission factor corresponds to the respective emissions of the three conditions.

SUPPLEMENTAL VII-1 and VII-2: Please use different colors for different species (ie.

use a single color for Cu and not for any other species). Please also increase the text size of axes titles and tick mark labels. RESPONSE: Different colours are used for different species and single colours for Cu, EC, Pyrene. The text size is increased.

SUPPLEMENTAL VIII: Please increase the text size of axes titles, tick mark labels, and other text. RESPONSE: The text size is increased.

SUPPLEMENTAL IX: Given the attention paid to ratios in this study, I would suggest adding rows for the most relevant ratios (at the least, Cu/Fn, Cu/Mn, and Cu/Sn) where data is available. RESPONSE: The most relevant ratios (Cu/Fe, Cu/Sb, Cu/Sn, Cu/Mn) are now in the table.

–EDITORIAL COMMENTS– PAGE1, LINE 13: grammar: add comma: "vehicular emissions, a large comprehensive dataset" RESPONSE: Now amended.

PAGE 1, LINE 20: ambiguous, replace "Most of the first ones" with "Light-duty traffic emission factors" and move "in absence of significant non-combustion emissions" to be beginning of the sentence (since it applies to the traffic emission factors, not the chassis dynamometer measurements) RESPONSE: The referee is right, it is corrected.

PAGE 1, LINE 21: suggested change to correct grammar and increase clarity: "Since recent measurements in Europe including those from this study are consistent, ratios involving copper (Cu/Fe and Cu/Sn) could be used as brake-wear emissions tracers as long as brakes with Cu remain in use." RESPONSE: the referee's proposal is accepted.

PAGE 1, LINE 23: The sentence regarding OC/EC ratio does not seem relevant or necessary to the abstract. In addition, the language implies that the OC/EC ratio is always 0.44 in France, which is likely not the intention of the authors. RESPONSE: The referee is right, the OC/EC ratio is not always 0.44 in France since it depends on the traffic fleet and other influential sources. The sentence is modified in the following way: "Near the Grenoble ring road, where the traffic was largely dominated by diesel

vehicles in 2011 (70 %), the OC/EC ratio estimated for traffic emissions was around 0.4."

PAGE 1, LINE 26: grammar: change "markers; while, their" to "markers, since" RESPONSE: "since" modifies what we attempted to mean, "but" will replace "while".

PAGE 1, LINE 28: grammar: "environments" (should be plural as written) RESPONSE: Modified

PAGE 1, LINE 30: grammar: "filters have been progressively introduced" RESPONSE: Modified

PAGE 1, LINE 36: soften tone: delete "It is obvious that" RESPONSE: Modified

PAGE 2, LINE 1: grammar: "Also, knowledge of the deleterious impacts of PM on human health" RESPONSE: Modified

PAGE 2, LINE 2: grammar: "PM is responsible" RESPONSE: Modified

PAGE 2, LINE 11: grammar: "They also do not represent the variability" RESPONSE: Modified

PAGE 2, LINE 19: clarity / word choice: replace "chemistry of PM" with "chemical composition of exhaust and non-exhaust particulate emissions" RESPONSE: That is a good proposal, it is modified.

PAGE 4, LINE 26: clarity: replace "below installation of" with "below this threshold for" RESPONSE: Modified

PAGE 4, LINE 28: grammar: "30% and 36% respectively" RESPONSE: the first "respectively" is removed.

PAGE 4, LINE 33: "(see SI section IV)" RESPONSE: "section IV" is added.

PAGE 5, LINE 12: grammar: replace "sampler" with "samples" RESPONSE: Modified

PAGE 5, LINE 13: ambiguity, makes it sound like there are two background and two

urban sites: delete "two" RESPONSE: Modified

PAGE 6, LINE 11: I am not sure if perhaps the authors intended "residual" instead of "residues"? RESPONSE: The authors mean "residues", it is amended.

PAGE 6, LINE 33: Is the discussion of the additional Fe and Mn source in the subsequent paragraph? I do not see it in Section 3.2 as indicated in the text. RESPONSE: This sentence is removed and the discussion on the additional source is page 7, lines 13-22.

PAGE 7, LINE 1: clarity / ambiguity: "are more scattered possibly due to differences between light-duty and heavy-duty emissions factors" RESPONSE: In this part, there is no conclusion on emission factors, but only on the strengths of relationships between datasets. The sentence is modified in order to be better understood (now page 7 lines 11-12). "Here, Cu, Fe and Sn are the metals that are the most closely related (Pearson $r2 \geq 0.8$), while relationships with Mn and Sb are more scattered (Pearson $r2 < 0.5$) and more closely related to the heavy-duty traffic (Table 1a)."

PAGE 7, LINE 12: precision / clarity: replace "relationships" with "correlations" RESPONSE: Modified

PAGE 7, LINES 33-34: Do the authors intend that the n-alkanes and hopanes are correlated to each (traffic, NOx, EC)? If so, change to "NOx, and EC". RESPONSE: No, we do not. The sentence is slightly modified as follows in order to avoid any ambiguity. "However, they are significantly correlated with NOx and EC and some of them with heavy-duty traffic." (page 8 lines 4-5)

PAGE 8, LINE 25: grammar: concentrations should be singular: "one for Sr concentration" RESPONSE: Modified

PAGE 8, LINE 36: grammar: "All of these suggest" RESPONSE: Modified

PAGE 9, LINES 8-9: To increase clarity, consider moving this sentence to the end of the first paragraph of this subsection (3.1.2). RESPONSE: Modified

PAGE 9, LINE 18: grammar: "contributes" RESPONSE: Corrected. The sentence is in part 2.4.

PAGE 9, LINE 27: change "technics" to "techniques" for more common spelling RE-SPONSE: Modified

PAGE 9, LINE 37: grammar: "larger than could be expected" RESPONSE: Modified

PAGE 10, LINE 2: grammar: "as ours, and the EF for exhaust OC" RESPONSE: Modified

PAGE 10, LINE 5: grammar: "contribution of smoker vehicles; and rapid formation" RESPONSE: Modified

PAGE 10 LINE 11-12: grammar: non-retrofitted is less clear than "without" (see comment on PAGE 15 LINE 6 as well): "test diesel vehicles without particle filters" RESPONSE: Modified

PAGE 10 LINE 26: grammar: "third highest traffic emission rate" RESPONSE: Modified

PAGE 11 LINE 28: grammar: "EF" RESPONSE: Modified

PAGE 11, LINE 30 – PAGE 12, LINE 8: The second sentence of this paragraph (regarding Cu and Sb brake wear emissions) confused me because it suggested that the Cu/Sb ratio would be a good candidate as a tracer, which is not the case (as communicated later in the paragraph). I suggest being more direct with the conclusion earlier in the paragraph to avoid confusion. RESPONSE: The second sentence is modified as follows in order to avoid any confusion. "[. . .] Cu/Sb ratio was often considered as a candidate to trace brake wear emissions"

PAGE 11 LINE 35: tone: suggestion to avoid the word "obviously." Also, please qualify the sentence by adding "in this study." RESPONSE: The statement is on the basis of national and European inventories. The first sentence is modified to be better understood. "According to inventories atmospheric copper is largely from brake wear." (p12, line 5)

PAGE 11 LINE 31: Add citation for CITEPA in references RESPONSE: the citation is added in references. CITEPA, édition mars 2018. Inventaire des émissions de polluants atmosphériques en France métropolitaine, format CEE-NU, https://www.citepa.org/images/III-1_Rapports_Inventaires/CEE-NU/UNECE_France_mars2018.pdf

PAGE 11 LINE 35: grammar: "depend" RESPONSE: modified

PAGE 12 LINE 6: replace "Then" with "Thus" RESPONSE: modified

PAGE 12 LINE 23: replace "spent" with "used" RESPONSE: modified

PAGE 12 LINE 31: precision / clarity: "Cu/Sn would be the strongest candidate RESPONSE: modified

PAGE 14 LINE 12: grammar / clarity: "The normalized abundance of $17\alpha,21\beta$norhopane (246 $\mu$g per g of OC) ..." RESPONSE: modified

PAGE 14 LINE 17: replace "data" with "emissions factors and compositions" RESPONSE: "data" is replaced by "EFs and normalized abundances" (p14, lines 31-32).

PAGE 15 LINE 2: grammar: "This study determined ...identified ...quantified" or "This study attempted to determine...identify...quantify" RESPONSE: Modified

PAGE 15 LINE 6: see comment on PAGE 10 LINES 11-12 for similar issue: "passenger diesel cars without particle filters" RESPONSE: Modified

PAGE 15, LINE 5: I am not sure what the authors are communicating with this first sentence ("The traffic shows the larger emission factor..."). Please rephrase. RESPONSE: Now rephrased (p15 line 20): "EC has the highest traffic emission factors and is strongly. . ."

PAGE 15 LINE 19: As written, implies that different sites in Europe were sampled in this study. Instead: "Cu/Fe ratios consistent with literature values from other sites suggest similar brake composition for these elements throughout Europe (as long as Cu-free brakes do not increase in use)." RESPONSE: The modification is accepted.

PAGE 15 LINES 21-23: Ambiguous as written as to whether these ratios are good or bad tracers. Instead: "Our measurements do not support the use of Cu/Mn and Cu/Sb as tracers of brake wear emissions possibly due to additional sources of Mn as well as the introduction of Sb-free brake pads." RESPONSE: Since the use of Cu/Mn and Cu/Sb cannot be strongly rejected by this study, I really prefer to replace the 2 sentences by: "Our measurements support more the use of Cu/Sn than that of Cu/Mn and Cu/Sb as tracers of brake wear emissions possibly due to additional sources of Mn and the introduction of Sb-free brake pads." (page 15-16 lines 38-1)

PAGE 15 LINE 36: clarity: "agreement between chassis dynamometer and near-road measurements" RESPONSE: To be as accurate as possible, the following is added: "between chassis dynamometer and near-road measurements and between this study and other recent studies." (page 16, lines 12-13)

PAGE 15 LINE 36: delete "the change of the" RESPONSE: Modified

PAGE 16 LINES 5-6: I suggest a more specific concluding sentence. Perhaps replace "delivered valuable information" with "describes exhaust and non-exhaust emissions measurements" RESPONSE: Taking into account the comments of both referees, I propose the following amendment (in the new manuscript, page 16, lines 20-21): "This study determines many quantitative data of traffic exhaust and non-exhaust emissions that could help in a better definition of traffic emissions in source apportionment studies."

Please also note the supplement to this comment:
https://www.atmos-chem-phys-discuss.net/acp-2018-816/acp-2018-816-AC1-

supplement.pdf

[revised manuscript text omitted]

Table I-2: Description of vehicle exhaust sampling: Sample flow rates, sampling time durations and dilution ratios. F1 is the filter dedicated to EC/OC analyses and F2 the filter dedicated to the other analyses. *: The selected flow rates were finally the same as the ones of the E3D vehicle. $: This vehicle has not been kept for the main vehicle tests.

**II. PM₁₀ mass and chemistry measured at the traffic site**

| | PM$_{10}$ | PM$_{2.5}$ | OC | EC | NO | NO$_2$ | NO$_x$ |
|---|---|---|---|---|---|---|---|
| *Unit* | $\mu g/m^3$ | $\mu g/m^3$ | $\mu g/m^3$ | $\mu g/m^3$ | $\mu g/m^3$ | $\mu g/m^3$ | $\mu 
[revised manuscript text omitted]

[Figure]

**e**

Figure VII-2: 4-hour concentrations measured at the traffic site: a: elemental carbon and pyrene; b: tricosane, tetracosane and 17α21βNorhopane; c: pyrene and phenanthrene

**VIII. Comparison with PM₁₀ emission factors of the recent literature (brake wear elements)**

| PM$_{10}$ EFs | This study | Johansson et al. 2009 | Bukowiecki et al. 2009$ | | Bukowiecki et al. 2009$§§$ | Gillies et al., 2001 | Handler et al. 2008 | Alves et al. 2015 | Hulskotte et al. 2004 | |
|---|---|---|---|---|---|---|---|---|---|---|
| Location | Grenoble, France | Stockholm, Sweden | Zürich, Switzerland | | Reiden Switzerland | L.A., U.S. | Vienna, Austria | Braga, Portugal | The Netherlands | |
| Road type | Urban freeway 4 lanes | Roadside | City centre Street canyon 2 lanes | | Inter urban freeway 4 lanes | Tunnel 2 bores with 3 lanes | Highway tunnel | Urban tunnel | Brake discs and pads analyses | |
| Traffic conditions | Mainly congested | Densely trafficked | Queues at red lights | | Mainly free-flowing | | Free flowing + congested | | | |
| Speed limit/inf. | 90 km.h$^{-1}$ | | 50 km.h$^{-1}$ | | 120 km.h$^{-1}$ | 42.6/64.4 km.h$^{-1}$ | 80 km.h$^{-1}$ | | | |
| Nb veh/day | 65-95,000 | | 22,000 | | 50,000 | 3000 veh/hr | 36-50,000 | 6,4-10,700 | Passenger cars Average brake profile | |
| %HDV | 0.3-12% | | 10%$ | | 15%$ | 2.6% | 4-12.6% | 10% | | |
| Results expected | Brake + resuspension | Brake + resuspension | Brake + resuspension | Brake only | Brake only | Brake + resuspension | Brake + resuspension | Brake + resuspension | Low braking 8 mg.veh$^{-1}$.km$^{-1}$ | High braking 15 mg.veh$^{-1}$.km$^{-1}$ |
| **Ba** µg.veh$^{-1}$.km$^{-1}$ | 66 | | 145 | 39.1 | 11.9 | 1040 | 55 | 670 | | |
| **Cr** µg.veh$^{-1}$.km$^{-1}$ | 43 | 41 | | | | 20 | | 60 | 30 | 55 |
| **Cu** µg.veh$^{-1}$.km$^{-1}$ | 300 | 542 | 476.6 | 108.1 | 28.2 | 530 | 156 | 110 | 291 | 546 |
| **Fe** mg.veh$^{-1}$.km$^{-1}$ | 6.71 | | 6.83 | 1.85 | 0.56 | 12.39 | 3.4 | 0.51 | 5.75 | 10.78 |
| **Mn** µg.veh$^{-1}$.km$^{-1}$ | 62 | 110 | | | | 70 | 42 | 60 | 43 | 80 |
| **Sb** µg.veh$^{-1}$.km$^{-1}$ | 27 | 144 | 74.1 | 17.9 | 32.3 | 220 | 100 | 50 | 64 | 120 |
| **Sn** µg.veh$^{-1}$.km$^{-1}$ | 55 | 126 | 72.5 | 16.1 | 8.7 | 70 | 25 | | 83 | 155 |
| **Ti** µg.veh$^{-1}$.km$^{-1}$ | 28 | | | | | 60 | 47 | 300 | 32 | 61 |
| **Cu/Fe*** | 0.046 ±0.015 | | 0.070 | 0.058 | 0.050 | 0.043 | 0.046 | 0.216 | 0.051 | 0.051 |
| **Cu/Sb*** | 12.6 ±4.7 | 3.8 | 6.4 | 6.0 | 0.9 | 2.4 | 1.6 | 2.2 | 4.5 | 4.6 |
| **Cu/Sn*** | 5.6 ±1.8 | 4.3 | 6.6 | 6.7 | 3.2 | 7.6 | 6.2 | | 3.5 | 3.5 |
| **Cu/Mn*** | 5.7 ±2.9 | 4.9 | | | | 7.6 | 3.7 | 1.8 | 6.8 | 6.8 |

$: traffic EFs related to an average 10% or 15% HDV and calculated from the three particle size fractions: 2.5-10; 1-2.5; and 0.1-1 µm for Zürich and the two size fractions 2.5-10 and 1-2.5 µm for Reiden; §: estimation for brake wear only. *The ratios are calculated from the published data.

---

## Author Comment (AC2) · 23 Jan 2019

The study by Charron et al. documents measurements of PM10 samples collected at a roadside in France. Overall, the results appear to be of high quality and are of potential use to further source apportionment work. The authors contextualize the study with the increasing importance of non-exhaust PM to total vehicular emissions, which is an interesting and intuitive concept. RESPONSE: The authors thank the reviewer for supporting their work.

Justiïfication for sampling PM10 should be included in the paper. What is the typical size distribution for vehicular emissions? Exhaust emissions, especially those treated

with a particulate filter on board the vehicle are dominated by ultrafine particle emissions, which have vanishingly small mass, especially in comparison to the remainder of PM10. Therefore, does PM10 over-sample the resuspended and non-exhaust emissions? Are aerosol size distributions of non-exhaust emissions known? The size-segregated composition of the PM in this study probably varies strongly, and will have an influence on the findings. Please discuss this aspect of the study in greater detail. Coarse particles also have fast deposition rates, limiting their influence on respiratory exposure compared to PM2.5 (or size classes with smaller upper size limits). RESPONSE: The reviewer is right, we can expect that the proportion of non-exhaust emissions in total traffic emissions is larger for PM10 than for PM2.5 or PM1. In this study, PM10 sampling was done in a deliberate manner to account for the entire breathable fraction of non-exhaust emissions. There are now certainties that PMcoarse have an impact on health (Beelen et al., 2014; Cheng et al., 2015; Malig et al., 2013), and the health of people living near traffic emissions can be strongly impacted. For that reason, we believe that the coarse fraction of particles emitted by traffic also needs consideration. Also PM10 in ambient air is also a quantity that is regulated in Europe and in many European traffic sites the European daily limit value is exceeded more than 10% of time. Note that during the same campaign, PM1 measurements were carried out using on-line instruments, results are published in another paper (DeWitt et al., 2015). The current state of the art of knowledge about the aerosol size distributions of non-exhaust emissions was the purpose of recent reviews (Thorpe and Harrison, 2008, Grigoratos and Martini, 2014), not of this work. But I agree that consideration of the different particle sizes could substantially improve knowledge on particulate traffic emissions. - I propose to first present and explain our choice in the introduction as follows: "(Recently, on the one hand, Shirmohammadi et al. (2017) and Weber et al., (2018) have shown the important role of non-tailpipe emissions to the oxidative potential of particulate matter species identified as tracers of vehicle abrasion; and on the other hand, the health impacts of coarse particles is now better documented (Beelen et al., 2014; Cheng et al., 2015; Malig et al., 2013). Therefore, a better knowledge on

vehicular emissions is required to better understand their contribution to urban atmospheric PM10 concentration levels and related health effects. [. . .] This study focuses on PM10 in order to take into account for the entire breathable fraction of non-exhaust particulate emissions, a large part of which are coarse particles (Thorpe and Harrison, 2008; Grigoratos and Martini, 2014)." (p2, line 7-8; lines 25-27 in the new version of the manuscript). - And then, to conclude on the need of research on different particle size fractions of non-exhaust vehicular emissions. "The determination of the particle size distribution of OC could improve knowledge of the organic emissions of traffic." (p 15 lines 28-29, new Ms) "Similarly to OC, the determination of the particle size distribution of metals may possibly improve the discrimination between influential sources in urban areas." (p 16 lines 5-7) Note that the new references are added.

Inclusion of more data visualizations throughout the manuscript, especially related the early sections of results (temporal profiles) is strongly advised. The reviewer's comprehension of the text was heavily improved by opening the SI, which should not be a requirement of reading a paper. This recommendation will likely improve the manuscript by an important margin. Combining metals onto fewer panels may help so that the entirety of SI Section VII does not need to be included as-is (for instance). Please consider this idea for all sections of the manuscript. RESPONSE: A part of the SI VII (temporal variations) is included in the paper as Figure 3 as well as the entire SI VIII (linear relationships) as Figure 4. Unfortunately, combining metals onto fewer panels reduces the readability, so this option has not been kept.

The presentation of the manuscript overall could use some English grammar editing and overall proof reading. A number of small grammatical or usage errors exist throughout the document. RESPONSE: An important correction work of grammatical errors has been done by reviewer 2 and the manuscript has been re-read.

A few salient ones are pointed out in the minor comments below, but this should not be considered an exhaustive list of corrections. Overall opinion: This study should be considered for publication after addressing the comments of this review in a major

revision. Introduction: The study is contextualized well, focused on the importance of nonexhaust emissions from vehicles, which may now be the most important aspect of vehicular emissions. Findings were, however, associated with both exhaust and nonexhaust emissions, which is clear. Methods: Methods were clearly described, with the exception of a lack of definition for TEOM-FDMS (a minor comment). RESPONSE: The TEOM-FDMS used during the field campaign are now defined: "PM10 and PM2.5 mass concentrations were also continuously measured using 8500C series TEOM-FDMS (Filter Dynamics Measurement System and Tapered Element Oscillating Microbalance mass sensor housed in a single-cabinet compact enclosure)."

Results: The use of the term 'incremental' is confusing (occurs throughout 'Results'). Referring to the "increment" with a clearer name would greatly clarify the presentation this important quantity. RESPONSE: in order to avoid any confusion, "incremental concentrations" is replaced by "increments in concentration" or "local increments in ... concentration due to traffic".

Section 3.1.1: It would be helpful to see a graphical representation of the contribution of each component of PM10 to the total mass. RESPONSE: A figure is added as Figure 2 and this graphical representation well highlights the differences between both sites.

Page 6, Line 32: It is becoming clear that the influence of particle size may be evident in the data (also see broader comments above). Please include a description or reference to what is known about the size-resolved composition of vehicle emissions from both exhaust and non-exhaust sources. [Such a discussion is not necessarily relevant to insert at this point in the manuscript, however, its importance began to become clear at this point.] RESPONSE: Of course, using another particle size for sampling could help with the separation of traffic metallic emissions and emission from the nearby industrial source, but it was not the purpose of the work. As suggested above, recommendations for further research are included in the conclusions: "Similarly to OC, the determination of the particle size distribution of metals may possibly improve the discrimination between influential sources in urban areas."

Page 7, line 38 – Page 7, line 2: Can a consideration for super-emitting vehicles be included in this part of the discussion? RESPONSE: It is now specified that none of tested vehicles was a high-emitting vehicle.

Page 8, lines 20-22: The measured 'dominant regional contribution' to OC may be masking the vehicular primary OC due to differences in the size distribution. Regional, secondary OC may be significantly aged, and therefore larger in size. While this will manifest as an overwhelming signal in a PM10 sample, it may not be so in a PM1 sample, or even a PM0.1 sample. Please discuss the significance of what is known about size-dependent composition of particles associated with vehicle emissions. RESPONSE: I agree with that secondary OC generates an overwhelming signal in PM10, but this is also the case for PM1 samples (see DeWitt et al., 2015). Additionally, the size distribution of traffic OC may possibly be more complex. Secondary OC may be larger in size than OC freshly emitted from the combustion chamber of the vehicles, but other traffic sources of OC (from tyre wear, road wear...) may also be larger in size than exhaust OC and possibly larger than secondary OC. While the examination of the size distribution of OC would be very valuable and requires further studies, considering only PM10 does not change the conclusion. I propose to add (p10, lines 17-18, after "Further studies are required to assess the respective importance of these processes."): "In particular, a better knowledge of particle size distribution of OC emitted by traffic might be useful."

Page 8, lines 33-38: If Co does not show any correlation with traffic emissions, how can it be reasonably concluded from the data collected in the present study that Co is a contributor to brake wear? The authors clearly attempt to make a case using the literature. If the contribution of other elements to one another is going to used to explain their similar vehicular sources (Fe, Cu, etc), then the same standard must be held to Co. At best, perhaps there is some other, more consistent, low level source of Co that is flattening the temporal profile. A conclusion as written in this passage, however, is dubious. RESPONSE: This is another example of overwhelming signal

from the background or possibly from another local source. Since concentrations of Co are significantly higher at the roadside site, other authors have made the same observation, and Co is measured in brake pads, we cannot totally exclude that Co is not emitted by the traffic. The end of text is changed as follows (page 9 lines 11-13): "All of these suggest that a possible contribution of brake wear to Co concentrations cannot be excluded, despite the lack of correlation with traffic indicators."

Page 9, lines 11-15: Can the impact of variability within the parts of the vehicle fleet be incorporated? In such highly controlled emissions from vehicles, a small number of super-emitters may be quite impactful. RESPONSE: Yes, the referee is right, the variability due to the presence of a few high emitting vehicles is now incorporated page 9 line 24. "(This variability reflects the presence of vehicles with various emission levels (diesel/petrol; different standards and engine load; cold start/hot vehicles;) presence of a few high emitting vehicles)."

Page 9, lines 34-35: This is a salient point for air quality management and regulation. RESPONSE: The referee is right. This point is added to the conclusion: "EC emissions from heavy-duty vehicles are estimated to be 5 times higher than those for light duty vehicles".

Page 10, line 5 (and other instances): Please give a more succinct deïnition to the term "smoker vehicles". Perhaps the authors could call these super-emitters? (see also comment about Page 9, lines 11-15) RESPONSE: "smoker vehicles" is replaced by "high emitting vehicles".

Page 11, line 22: The authors use and cite a ïnding that a dominant fraction of brake wear emissions come from the disc and not the brake pad. This seems hard to believe considering the fact that the brake pad is the primarily consumable part, and that brake discs (rotors) do not need to be replaced as often. RESPONSE: That is a judicious remark of the referee. This assumption is based on the work of Hulskotte et al., 2014 and this sensible point is already discussed in Hulkotte's et al. (2014) that

draw on research from Sander et al. (2003) and Varrica et al. (2013), I quote: "it can be concluded that a share of 30%±5% of brake pad wear within total brake wear probably is a realistic value. This in a reasonable agreement with Sander et al. (2003) who reported that at low-metallic brakes (as most frequently used in Europe) about 60% of mass loss has his origin in the brake disc [here we estimated that 70% comes from discs]. While this is a surprisingly result at first, a closer look at a spent brake pad and disc reveals that there can be rather deep wear patterns on the disc and, most important, the surface area of the disc is much larger. So, while the wear in mm from a brake pad might be higher, once this is multiplied with surface area, the wear from the discs appears to be larger. None of the spent brake pad collected was worn so much that the iron base of the brake pad in any way. We assume that in the real world abrasion of the iron base of the brake pad very rarely will occur because obliged periodical roadworthiness tests will prevent this situation. So we may assume that wear of the iron brake pad base will not contribute to emission of iron. Recently Varrica et al. (2013) found that half the concentration of antimony in brake pad residue on wheel rims compared to brake pad linings, suggesting a 50% contribution by brake discs" Since at this state of knowledge the influence of unknown parameters on the high iron emissions observed at different locations cannot be excluded, and new researches are needed anyway, I added in the text (page 11 lines 34-36): "This latter assumption is supported by other researches (Sander et al., 2003; Varrica et al., 2013), even though the generation mechanisms of brake wear particles have not been fully understood yet (Grigoratos and Martini, 2014)."

Page 11, line 35: Please clarify and/or define "PM fraction". This term is not used routinely in this manuscript. A change in wording may help in this instance. RESPONSE: "PM fraction" is replaced by "PM size fraction".

Page 11, line 34 - Page 12, line 8: It may help to define Cu/Sb in the brake materials themselves. How might consistency in the brake materials themselves drive atmospheric Cu/Sb? Could the act of particle formation (temperature, breaking force,

etc) influence the ratio? RESPONSE: These are excellent questions and a field of research to explore which I cannot answer. Not enough is known on the Cu/Sb in the brake materials leading to airborne particles and a few researches are cited in the text. A recent review already exist (Pant and Harrison, 2013) and is already cited in the text. Also, as noted above, little is known on the generation mechanisms of brake wear particles and how the generation mechanisms influence the physicochemical characteristics of atmospheric brake wear particles. Moreover published works show a wide variety of sampling methodologies that lead the results difficult to compare.

Page 15, line 17-19: Has Cu/Fe been reported in any other proportion in the atmosphere? Please illustrate (perhaps in the results section on this topic) that the âĹij4% value is unique to vehicles. RESPONSE: Data measured at the urban background Les Frênes, unpublished data from other French sites and published data at other sites (e.g. Hueglin et al., 2005) show that the Cu/Fe ratios are different in rural and urban background areas (sometimes similar in urban background sites strongly influenced by traffic). This part is modified as follows (page 15, lines 36-39): "Cu/Fe ratios in agreement with literature values for other kerbside sites, while Cu/Fe ratios may be different for urban background or rural sites (e.g. Hueglin et al., 2005; this study: Les Frênes' data), suggest similar brake composition for these elements throughout Europe (as long as Cu-free brakes do not increase in use)."

Page 15, line 24-28: This is a strange placement for an overview paragraph about the significance of redox-active metals, which have only been mentioned as such in the introduction. This paragraph is probably better off at the beginning of the conclusions section, mirroring the structure of the paper itself. RESPONSE: The paragraph is removed and references are added in the introduction (to complete the discussion on the health effect of redox-active metals). "There is now strong evidence that traffic-related PM is responsible for adverse health effects due to the health effect of both carbonaceous material from exhaust emissions and redox-active metals in traffic-generated dust including road, brake and tyre wear (Kukutschová et al., 2009; Cassee et al.,

2013; Amato et al., 2014 and references wherein; Pardo et al., 2015; Poprac et al., 2017). Recently, on the one hand, Shirmohammadi et al. (2017) and Weber et al., (2018) have shown the important role of non-tailpipe emissions to the oxidative potential of particulate matter species identified as tracers of vehicle abrasion;[. . .]"

Page 15, line 29: While studies such as this one may be scarce in the literature, the authors have reported agreement with these studies throughout the manuscript – suggesting that the science is highly convergent. Please provide a clear, summary assessment of the novel findings of this study. RESPONSE: The paragraph now begins with (page 16 lines 4-6): "Particulate organic emission data for European motor vehicles is scarce. In this study, a few PAHs, n-alkanes and hopanes have been identified as organic molecular markers of fresh diesel traffic emissions and their emission factors have been quantified."

Page 16, lines 5-6: This concluding statement seems to highlight the fact that this is a characterization study with little in the way of entirely new findings. (see previous comment) Do the authors believe feel that this is the case? If not, a revised summary statement or declaration of a way forward in light of the present study is in order. RESPONSE: Taking into account the comments of both referees, I propose the following amendment (page 16 lines 20-21): "This study determines many quantitative data of traffic exhaust and non-exhaust emissions that could help in a better definition of traffic emissions in source apportionment studies." Considering the two comments above, the first paragraph of the conclusion is rewritten in order to better show the most important points and novel findings related to this research (very large dataset including metal, major ions and organic / both in situ and chassis dynamometer measurements / not only identification but also quantification of tracers that could be used in source apportionment studies) (page 15 lines 13-17): "Thanks to a very large comprehensive dataset of particulate species collected from a simultaneous near-road and urban background measurement field campaign and chassis dynamometer experiments of a few in-use passenger cars, this study was able to determine emission factors for many

particulate species from road traffic and to identify and quantify tracers of exhaust and non-exhaust vehicular emissions that could be used in source apportionment studies."

Minor comments: Line 20-21: "Most of the first ones" – please be specific, most of the first 'what'? What do you mean by first? I honestly do not know to which prior items this sentence refers. RESPONSE: It is replaced by "In absence of significant non-combustion emissions..." (page 1 line 20)

Line 24-26: "On the contrary,..." to what?? RESPONSE: It is replaced by "Although..." (line 25)

Page 8, line 9: should refer to "SI Section VII" RESPONSE: It is amended.

Page 9, line 27: change "technics" to "techniques" RESPONSE: it is amended.

Page 10, line 2: define DPF, first usage RESPONSE: it is replaced by Diesel Particulate Filter.

Page 10, line 26: change to "third highest" RESPONSE: Now, the sentence is (page 10 line 37): "Fe presents by far the third highest traffic emission rate after those of EC and OC"

Please also note the supplement to this comment:
https://www.atmos-chem-phys-discuss.net/acp-2018-816/acp-2018-816-AC2-supplement.pdf
* * *
[Figure]

Figure 2: Median concentrations measured at the roadside site (Echirolles) and urban background site (Les Frênes) and comparison with respective median TEOM-FDMS PM$_{10}$ concentrations. OM is computed using the factor 1.8 estimated for Grenoble city (Favez et al., 2010).

**Fig. 1.**

**Supplement:**

**Identification and quantification of particulate tracers of exhaust and non-exhaust vehicle emissions**

Aurélie Charron[1,2], Lucie Polo-Rehn[1,2], Jean-Luc Besombes[3], Benjamin Golly[3], Christine Buisson[4], Hervé Chanut[5], Nicolas Marchand[6], Géraldine Guillaud[5], and Jean-Luc Jaffrezo[1]

**Supplementary Information**

**I.    Characteristics of test vehicles and vehicle exhaust sampling**

The selected vehicles represent the most frequent vehicle classes in the French fleet in circulation, according to the following criteria: European emission standards (EURO classes), motorization (diesel or petrol), engine capacity (large-, intermediate-, small-engined cars, defined as vehicles with the following capacities: below 1.4 L, from 1.4 to 2 L, above 2 L), and the presence or not of an after-treatment system. According to André et al. (2014), Euro 3 and Euro 4 vehicles accounted in 2011 for the largest part of the French fleet in circulation, followed by Euro 5 and Euro 2 vehicles. They also estimated that diesel vehicles with medium engine displacement (1.4 to 2.0 liters) accounted for 78 % of the diesel vehicles, while petrol vehicles with small displacement (less than 1.4 liters) were the most numerous (62 % of petrol vehicles). Chassis dynamometer experiments have been conducted before the field campaign, and the small differences between the French fleet and the local fleet are discussed in Fallah Shorshani et al. (2015). Finally, three diesel and one petrol vehicles have been chosen as a good representation of the French fleet in circulation in 2011, as follows:

– Petrol Euro 2 with small engine displacement (< 1.4 l) – below referred as E2P

– Petrol Euro 4 with small engine displacement (< 1.4 l) – below referred as E4P

– Diesel Euro 3 with medium engine displacement (1.4 – 2 l) – E3D

– Diesel Euro 4 with medium engine displacement (1.4 – 2 l) - E4D

– Diesel Euro 4 with medium engine displacement (1.4 – 2 l) and equipped with a particulate filter, instead a Euro 5 diesel vehicle) – referred as E4D-PF. Note that this vehicle met Euro 5 standard for PM emissions thanks to the presence of the particle filter, but not the Euro 5 standard for NOx emissions.

No high-emitting vehicles has been selected (all test vehicles followed their own EU regulations).

| vehicles | Vehicle fuel | EURO class | Vehicle model | Year | Mileage km | Engine dCi | Emission control device |
|----------|-------------|-----------|---------------|------|-----------|-----------|-------------------------|
| E4P | Petrol | EURO 4 | Renault Clio 3 | 2006 | 82,000 | 1.4 | TWC |
| E2P | Petrol | EURO 2 | Ford Ka | 1999 | 72,000 | 1.3 | CC |
| E3D | Diesel | EURO 3 | Xsara Picasso HDI | 2003 | 140,000 | 1.9 | DOC |
| E4D | Diesel | EURO 4 | Renault Kangoo | 2005 | 146,000 | 1.5 | DOC |
| E4D-PF | Diesel | EURO 4 | Audi TDI | 2009 | 73,800 | 1.9 | DOC+PF |

Table I-1: Detailed characteristics of vehicles run on chassis dynamometer. TWC: three-way catalyst; DOC: diesel oxidation catalyst; PF: particle filter; CC: catalyst converter

| vehicles | Flow rates (l.min-1) | | Sampling duration (min, nb cycles) | Mean dilution ratios | |
|----------|------|------|-----------------------------------|------------|------------|
| | F1 | F2 | | Road cycle | Urban cycle |
| E3D | 5 | 40 | 15 min (1 cycle) for F1 30 min (2 cycles) for F2 | 22 | 41 |
| E4D | 5-10* | 30-40-50* | 15 min (1 cycle) for F1 30 min (2 cycles) for F2 | 23 | 37 |
| E4D-PF | 30 | 50 | 30 min (2 cycles) | 18 | 28 |
| E2P | 30 | 50 | 30 min (2 cycles) | 17 | 28 |
| E4P$ | 50 | - | 30 min (2 cycles) | - | 23 |

Table I-2: Description of vehicle exhaust sampling: Sample flow rates, sampling time durations and dilution ratios. F1 is the filter dedicated to EC/OC analyses and F2 the filter dedicated to the other analyses. *: The selected flow rates were finally the same as the ones of the E3D vehicle. $: This vehicle has not been kept for the main vehicle tests.

**II. PM₁₀ mass and chemistry measured at the traffic site**

| | PM$_{10}$ | PM$_{2.5}$ | OC | EC | NO | NO$_2$ | NO$_x$ |
|---|---|---|---|---|---|---|---|
| *Unit* | $\mu g/m^3$ | $\mu g/m^3$ | $\mu g/m^3$ | $\mu g/m^3$ | $\mu g/m^3$ | $\mu g/m^3$ | $\mu 
[revised manuscript text omitted]

[Figure]

**e**

Figure VII-2: 4-hour concentrations measured at the traffic site: a: elemental carbon and pyrene; b: tricosane, tetracosane and 17α21βNorhopane; c: pyrene and phenanthrene

**VIII. Comparison with PM₁₀ emission factors of the recent literature (brake wear elements)**

| PM$_{10}$ EFs | This study | Johansson et al. 2009 | Bukowiecki et al. 2009$ | | Bukowiecki et al. 2009$§§$ | Gillies et al., 2001 | Handler et al. 2008 | Alves et al. 2015 | Hulskotte et al. 2004 | |
|---|---|---|---|---|---|---|---|---|---|---|
| Location | Grenoble, France | Stockholm, Sweden | Zürich, Switzerland | | Reiden Switzerland | L.A., U.S. | Vienna, Austria | Braga, Portugal | The Netherlands | |
| Road type | Urban freeway 4 lanes | Roadside | City centre Street canyon 2 lanes | | Inter urban freeway 4 lanes | Tunnel 2 bores with 3 lanes | Highway tunnel | Urban tunnel | Brake discs and pads analyses | |
| Traffic conditions | Mainly congested | Densely trafficked | Queues at red lights | | Mainly free-flowing | | Free flowing + congested | | | |
| Speed limit/inf. | 90 km.h$^{-1}$ | | 50 km.h$^{-1}$ | | 120 km.h$^{-1}$ | 42.6/64.4 km.h$^{-1}$ | 80 km.h$^{-1}$ | | | |
| Nb veh/day | 65-95,000 | | 22,000 | | 50,000 | 3000 veh/hr | 36-50,000 | 6,4-10,700 | Passenger cars Average brake profile | |
| %HDV | 0.3-12% | | 10%$ | | 15%$ | 2.6% | 4-12.6% | 10% | | |
| Results expected | Brake + resuspension | Brake + resuspension | Brake + resuspension | Brake only | Brake only | Brake + resuspension | Brake + resuspension | Brake + resuspension | Low braking 8 mg.veh$^{-1}$.km$^{-1}$ | High braking 15 mg.veh$^{-1}$.km$^{-1}$ |
| **Ba** µg.veh$^{-1}$.km$^{-1}$ | 66 | | 145 | 39.1 | 11.9 | 1040 | 55 | 670 | | |
| **Cr** µg.veh$^{-1}$.km$^{-1}$ | 43 | 41 | | | | 20 | | 60 | 30 | 55 |
| **Cu** µg.veh$^{-1}$.km$^{-1}$ | 300 | 542 | 476.6 | 108.1 | 28.2 | 530 | 156 | 110 | 291 | 546 |
| **Fe** mg.veh$^{-1}$.km$^{-1}$ | 6.71 | | 6.83 | 1.85 | 0.56 | 12.39 | 3.4 | 0.51 | 5.75 | 10.78 |
| **Mn** µg.veh$^{-1}$.km$^{-1}$ | 62 | 110 | | | | 70 | 42 | 60 | 43 | 80 |
| **Sb** µg.veh$^{-1}$.km$^{-1}$ | 27 | 144 | 74.1 | 17.9 | 32.3 | 220 | 100 | 50 | 64 | 120 |
| **Sn** µg.veh$^{-1}$.km$^{-1}$ | 55 | 126 | 72.5 | 16.1 | 8.7 | 70 | 25 | | 83 | 155 |
| **Ti** µg.veh$^{-1}$.km$^{-1}$ | 28 | | | | | 60 | 47 | 300 | 32 | 61 |
| **Cu/Fe*** | 0.046 ±0.015 | | 0.070 | 0.058 | 0.050 | 0.043 | 0.046 | 0.216 | 0.051 | 0.051 |
| **Cu/Sb*** | 12.6 ±4.7 | 3.8 | 6.4 | 6.0 | 0.9 | 2.4 | 1.6 | 2.2 | 4.5 | 4.6 |
| **Cu/Sn*** | 5.6 ±1.8 | 4.3 | 6.6 | 6.7 | 3.2 | 7.6 | 6.2 | | 3.5 | 3.5 |
| **Cu/Mn*** | 5.7 ±2.9 | 4.9 | | | | 7.6 | 3.7 | 1.8 | 6.8 | 6.8 |

$: traffic EFs related to an average 10% or 15% HDV and calculated from the three particle size fractions: 2.5-10; 1-2.5; and 0.1-1 µm for Zürich and the two size fractions 2.5-10 and 1-2.5 µm for Reiden; §: estimation for brake wear only. *The ratios are calculated from the published data.